# Consciousness-Inspired Spatio-Temporal Abstractions for Better Generalization in Reinforcement Learning

**Mingde Zhao**[1,3], **Safa Alver**[1,3], **Harm van Seijen**[4], **Romain Laroche,**
**Doina Precup**[1,3,5], **Yoshua Bengio**[2,3]
[*1]McGill University, [2]Université de Montréal, [3]Mila, [4]Sony AI, [5]Google DeepMind
{mingde.zhao,safa.alver}@mail.mcgill.ca, harm.vanseijen@sony.com
romain.laroche@gmail.com, dprecup@cs.mcgill.ca, yoshua.bengio@mila.quebec

## Abstract

Inspired by human conscious planning, we propose **Skipper**, a model-based reinforcement learning framework utilizing spatio-temporal abstractions to generalize better in novel situations. It automatically decomposes the given task into smaller, more manageable subtasks, and thus enables sparse decision-making and focused computation on the relevant parts of the environment. The decomposition relies on the extraction of an abstracted proxy problem represented as a directed graph, in which vertices and edges are learned end-to-end from hindsight. Our theoretical analyses provide performance guarantees under appropriate assumptions and establish where our approach is expected to be helpful. Generalization-focused experiments validate **Skipper**'s significant advantage in zero-shot generalization, compared to some existing state-of-the-art hierarchical planning methods.

## 1 Introduction

Attending to relevant aspects in both time and space, human conscious planning breaks down long-horizon tasks into more manageable steps, each of which can be narrowed down further. Stemming from consciousness in the first sense (C1) (Dehaene et al., 2020), this type of planning focuses attention on mostly the important decision points (Sutton et al., 1999) and relevant environmental factors linking the decision points (Tang et al., 2020), thus *operating abstractly both in time and in space*. In contrast, existing Reinforcement Learning (RL) agents either operate solely based on intuition (model-free methods) or are limited to reasoning over mostly relatively shortsighted plans (model-based methods) Kahneman (2017). The intrinsic limitations constrain the real-world application of RL under a glass ceiling formed by challenges of generalization.

Building on our previous work on conscious planning (Zhao et al., 2021), we take inspirations to develop a planning agent that automatically decomposes the complex task at hand into smaller subtasks, by constructing abstract "proxy" problems. A proxy problem is represented as a graph where 1) the vertices consist of states imagined by a generative model, corresponding to sparse decision points; and 2) the edges, which define temporally-extended transitions, are constructed by focusing on a small amount of relevant information from the states, using an attention mechanism. Once a proxy problem is constructed and the agent solves it to form a plan, each of the edges defines a new sub-problem, on which the agent will focus next. This divide-and-conquer strategy allows constructing partial solutions that generalize better to new situations, while also giving the agent flexibility to construct abstractions necessary for the problem at hand. Our theoretical analyses establish guarantees on the quality of the solution to the overall problem.

We also examine empirically advantages of out-of-training-distribution generalization of our method after using only a few training tasks. We show through detailed controlled experiments that the proposed framework, named **Skipper**, in most cases performs significantly better in terms of zero-shot generalization, compared to the baselines and to some state-of-the-art Hierarchical Planning (HP) methods (Nasiriany et al., 2019; Hafner et al., 2022).

---

[*]Work largely done during Mingde, Harm and Romain's time at Microsoft Research Montreal. Source code of experiments available at https://github.com/mila-iqia/Skipper

## 2 PRELIMINARIES

**Reinforcement Learning & Problem Setting.** An RL agent interacts with an environment via a sequence of actions to maximize its cumulative reward. The interaction is usually modeled as a Markov Decision Process (MDP) $\mathcal{M} \equiv \langle \mathcal{S}, \mathcal{A}, P, R, d, \gamma \rangle$, where $\mathcal{S}$ and $\mathcal{A}$ are the set of states and actions, $P : \mathcal{S} \times \mathcal{A} \to \text{Dist}(\mathcal{S})$ is the state transition function, $R : \mathcal{S} \times \mathcal{A} \times \mathcal{S} \to \mathbb{R}$ is the reward function, $d : \mathcal{S} \to \text{Dist}(\mathcal{S})$ is the initial state distribution, and $\gamma \in [0, 1]$ is the discount factor. The agent needs to learn a policy $\pi : \mathcal{S} \to \text{Dist}(\mathcal{A})$ that maximizes the value of the states, *i.e.* the expected discounted cumulative reward $\mathbb{E}_{\pi,P}[\sum_{t=0}^{T_\perp} \gamma^t R(S_t, A_t, S_{t+1})|S_0 \sim d]$, where $T_\perp$ denotes the time step at which the episode terminates. A value estimator $Q : \mathcal{S} \times \mathcal{A} \to \mathbb{R}$ can be used to guide the search for a good policy. However, real-world problems are typically partially observable, meaning that at each time step $t$, instead of states, the agent receives an observation $x_{t+1} \in \mathcal{X}$, where $\mathcal{X}$ is the observation space. The agent then needs to infer the state from the sequence of observations, usually with a state encoder.

One important goal of RL is to achieve high (generalization) performance on evaluation tasks after learning from a limited number of training tasks, where the evaluation and training distributions may differ; for instance, a policy for a robot may need to be trained in a simulated environment for safety reasons, but would need to be deployed on a physical device, a setting called sim2real. Discrepancy between task distributions is often recognized as a major reason why RL agents are yet to be applied pervasively in the real world (Igl et al., 2019). To address this issue, in this paper, agents are trained on a small set of fixed training tasks, then evaluated in unseen tasks, where there are environmental variations, but the core strategies needed to finish the task remain consistent. To generalize well, the agents need to build learned skills which capture the consistent knowledge across tasks.

**Deep Model-based RL.** Deep model-based RL uses predictive or generative models to guide the search for policies (Silver et al., 2017). In terms of generalization, rich models, expressed by Neural Networks (NNs), may capture generalizable information and infer latent causal structure. *Background* planning agents *e.g.*, Dreamer (Hafner et al., 2023) use a model as a data generator to improve the value estimators and policies, which executes in background without directly engaging in the environment (Sutton, 1991). These agents do not improve on the trained policy at decision time. In contrast, *decision-time* planning agents *e.g.*, MuZero (Schrittwieser et al., 2020) and PlaNet (Hafner et al., 2019) actively use models at decision time to make better decisions. Recently, Alver & Precup (2022) suggests that the latter approach provides better generalization, aligning with observations from cognitive behaviors (Momennejad et al., 2017).

**Options & Goal-Conditioned RL.** Temporal abstraction allows agents to use sub-policies, and to model the environment over extended time scales, to achieve both better generalization and the divide and conquer of larger problems. Options and their models provide a formalism for temporal abstraction in RL (Sutton et al., 1999). Each option consists of an initiation condition, a policy, and a termination condition. For any set of options defined on an MDP, the decision process that selects only among those options, executing each to termination, is a Semi-MDP (SMDP) (Sutton et al., 1999; Puterman, 2014), consisting of the set of states $\mathcal{S}$, the set of options $\mathcal{O}$, and for each state-option pair, an expected return, and a joint distribution of the next state and transit time. In this paper, we focus on goal-conditioned options, where the initiation set covers the whole state space $\mathcal{S}$. Each such option is a tuple $o = \langle \pi, \beta \rangle$, where $\pi : \mathcal{S} \to \text{Dist}(\mathcal{A})$ is the (intra-)option policy and $\beta : \mathcal{S} \to \{0, 1\}$ indicates when a goal state is reached. Hindsight Experience Replay (HER) (Andrychowicz et al., 2017) is often used to train goal-conditioned options by sampling a transition $\langle x_t, a_t, r_{t+1}, x_{t+1} \rangle$ together with an additional observation $x^{\odot}$ from the same trajectory, which is re-labelled as a "goal".

## 3 **Skipper**: SPATIALLY & TEMPORALLY ABSTRACT PLANNING

In this section, we describe the main ingredients of **Skipper** - a framework that formulates a **proxy** problem for a given task, solves this problem, and then proceeds to "fill in" the details of the plan.

### 3.1 PROXY PROBLEMS

Proxy problems are finite graphs constructed at decision-time, whose vertices are states and whose directed edges estimate transitions between the vertices, as shown in Fig. 1. We call the states

selected to be vertices of the proxy problems *checkpoints*, to differentiate from other uninvolved states. The current state is always included as one of the vertices. The checkpoints are proposed by a generative model and represent some states that the agent might experience in the current episode, often denoted as $S^\odot$ in this paper. Each edge is annotated with estimates of the cumulative discount and reward associated with the transition between the connected checkpoints; these estimates are learned over the **relevant** aspects of the environment and **depend** on the agent's capability. As the low-level policy implementing checkpoint transitions improves, the edges strengthen. Planning in a proxy problem is temporally abstract, since the checkpoints act as sparse decision points. Estimating each checkpoint transition is spatially abstract, as an option corresponding to such a task would base its decisions only on some aspects of the environment state (Bengio, 2017; Konidaris & Barto, 2009), to improve generalization as well as computational efficiency (Zhao et al., 2021).

A proxy problem can be viewed as a deterministic SMDP, where each directed edge is implemented as a checkpoint-conditioned option. It can be fully described by the discount and reward matrices, $\Gamma^\pi$ and $V^\pi$, where $\gamma^\pi_{ij}$ and $v^\pi_{ij}$ are defined as:

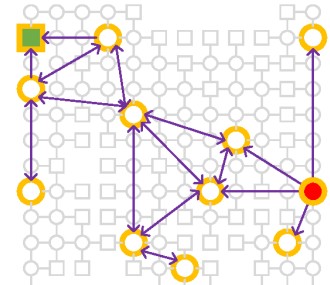

$$\gamma^\pi_{ij} \doteq \mathbb{E}_\pi \left[ \gamma^{T_\perp} | S_0 = s_i, S_{T_\perp} = s_j \right] \qquad (1)$$

$$v^\pi_{ij} \doteq \mathbb{E}_\pi \left[ \sum_{t=0}^{T_\perp} \gamma^t R_t | S_0 = s_i, S_{T_\perp} = s_j \right]. \qquad (2)$$

By planning with $\Gamma^\pi$ and $V^\pi$, *e.g.* using SMDP value iteration (Sutton et al., 1999), we can solve the proxy problem, and form a jumpy plan to travel between states in the original problem. If the proxy problems can be estimated well, the obtained solution will be of good quality, as established in the following theorem:

Figure 1: **A Proxy Problem on a Grid-World Navigation Task**: the MDP of the original problem is in gray and the terminal states are marked with squares. An agent needs to get from the (filled red) position, to the goal (filled green). Distant goals can be reached by leveraging a proxy problem with 12 checkpoints (outlined orange).

**Theorem 1** *Let $\mu$ be the SMDP policy (high-level) and $\pi$ be the low-level policy. Let $\hat{V}^\pi$ and $\hat{\Gamma}^\pi$ denote learned estimates of the SMDP model. If the estimation accuracy satisfies:*

$$|v^\pi_{ij} - \hat{v}^\pi_{ij}| < \epsilon_v v_{max} \ll (1-\gamma)v_{max} \qquad \textbf{\textit{and}} \qquad (3)$$

$$|\gamma^\pi_{ij} - \hat{\gamma}^\pi_{ij}| < \epsilon_\gamma \ll (1-\gamma)^2 \qquad \forall i, j.$$

*Then, the estimated value of the composite $\hat{v}_{\mu\circ\pi}(s)$ is accurate up to error terms linear in $\epsilon_v$ and $\epsilon_\gamma$:*

$$\hat{v}_{\mu\circ\pi}(s) \doteq \sum_{k=0}^\infty \hat{v}_\pi(s^\odot_k | s^\odot_{k+1}) \prod_{\ell=0}^{k-1} \hat{\gamma}_\pi(s^\odot_\ell | s^\odot_{\ell+1}) = v_{\mu\circ\pi}(s) \pm \frac{\epsilon_v v_{max}}{1-\gamma} \pm \frac{\epsilon_\gamma v_{max}}{(1-\gamma)^2} + o(\epsilon_v + \epsilon_\gamma)$$

*where $\hat{v}_\pi(s_i | s_j) \equiv \hat{v}^\pi_{ij}$ and $\hat{\gamma}_\pi(s_i | s_j) \equiv \hat{\gamma}^\pi_{ij}$, and $v_{max}$ denotes the maximum value.*

The theorem indicates that once the agent achieves high accuracy estimation of the model for the proxy problem and a near-optimal lower-level policy $\pi$, it converges toward optimal performance (proof in Appendix D.2). The theorem also makes no assumption on $\pi$, since it would likely be difficult to learn a good $\pi$ for far away targets. Despite the theorem's generality, in the experiments, we limit ourselves to navigation tasks with sparse rewards for reaching goals, where the goals are included as permanent vertices in the proxy problems. This is a case where the accuracy assumption can be met non-trivially, *i.e.*, while avoiding degenerate proxy problems whose edges involve no rewards. Following Thm. 1, we train estimators for $v_\pi$ and $\gamma_\pi$ and refer to this as *edge estimation*.

## 3.2 DESIGN CHOICES

To implement planning over proxy problems, **Skipper** embraces the following design choices:

**Decision-time planning** is employed due to its ability to improve the policy in novel situations;

**Spatio-temporal abstraction**: temporal abstraction breaks down the given task into smaller ones, while spatial abstraction[1] over the state features improves local learning and generalization;

---

[1]We use "spatial abstraction" to denote specifically the behavior of constraining decision-making to the relevant environmental factors during an option. Please check Section 4 for discussions and more details.

**Higher quality proxies**: we introduce pruning techniques to improve the quality of proxy problems;

**Learning end-to-end from hindsight, off-policy**: to maximize sample efficiency and the ease of training, we propose to use auxiliary (off-)policy methods for edge estimation, and learn a context-conditioned checkpoint generation, both from hindsight experience replay;

**Delusion suppression**: we propose a delusion suppression technique to minimize the behavior of chasing non-existent outcomes. This is done by exposing edge estimation to imagined targets that would otherwise not exist in experience.

### 3.3 PROBLEM 1: EDGE ESTIMATION

First, we discuss how to estimate the edges of the proxy problem, given a set of already generated checkpoints. Inspired by conscious information processing in brains (Dehaene et al., 2020) and existing approach in Sylvain et al. (2019), we introduce a local perceptive field selector, $\sigma$, consisting of an attention bottleneck that (soft-)selects the top-$k$ local segments of the full state (*e.g.* a feature map by a typical convolutional encoder); all segments of the state compete for the $k$ attention slots, *i.e.* irrelevant aspects of state are discouraged or discarded, to form a partial state representation (Mott et al., 2019; Tang et al., 2020; Zhao et al., 2021; Alver & Precup, 2023). We provide an example in Fig. 2 (see purple parts). Through $\sigma$, the auxiliary estimators, to be discussed soon, force the bottleneck mechanism to promote aspects relevant to the local estimation of connections between the checkpoints. The rewards and discounts are then estimated from the partial state $\sigma(S)$, conditioned on the agent's policy.

#### 3.3.1 BASIS FOR CONNECTIONS: CHECKPOINT-ACHIEVING POLICY

The low-level policy $\pi$ maximizes an intrinsic reward, *s.t.* the target checkpoint $S^\odot$ can be reached. The choice of intrinsic reward is flexible; for example, one could use a reward of $+1$ when $S_{t+1}$ is within a small radius of $S^\odot$ according to some distance metric, or use reward-respecting intrinsic rewards that enable more sophisticated behaviors, as in (Sutton et al., 2022). In the following, for simplicity, we will denote the checkpoint-achievement condition with equality: $S_{t+1} = S^\odot$.

#### 3.3.2 ESTIMATE CONNECTIONS

We learn the connection estimates with auxiliary reward signals that are designed to be not task-specific (Zhao et al., 2019). These estimates are learned using C51-style distributional RL, where the output of each estimator takes the form of a histogram over scalar support (Dabney et al., 2018).

**Cumulative Reward.** The cumulative discounted task reward $v_{ij}^\pi$ is learned by policy evaluation on an auxiliary reward that is the same as the original task reward everywhere except when reaching the target. Given a hindsight sample $\langle x_t, a_t, r_{t+1}, x_{t+1}, x^\odot \rangle$ and the corresponding encoded sample $\langle s_t, a_t, r_{t+1}, s_{t+1}, s^\odot \rangle$, we train $V_\pi$ with KL-divergence as follows:

$$\hat{v}_\pi(\sigma(s_t), a_t | \sigma(s^\odot)) \leftarrow \begin{cases} R(s_t, a_t, s_{t+1}) + \gamma \hat{v}_\pi(\sigma(s_{t+1}), a_{t+1} | \sigma(s^\odot)) & \text{if } s_{t+1} \neq s^\odot \\ R(s_t, a_t, s_{t+1}) & \text{if } s_{t+1} = s^\odot \end{cases} \quad (4)$$

where $\sigma(s)$ is the spatially-abstracted from the full state $s$ and $a_{t+1} \sim \pi(\cdot | \sigma(s_{t+1}), \sigma(s^\odot))$.

**Cumulative Distances / Discounts.** With C51 and uniform supports, the cumulative discount leading to $s_\odot$ under $\pi$ is unfortunately more difficult to learn than $V_\pi$, since the prediction would be heavily skewed towards 1 if $\gamma \approx 1$. Yet, we can instead effectively estimate cumulative (truncated) distances (or trajectory length) under $\pi$. Such distances can be learned with policy evaluation, where the auxiliary reward is $+1$ on every transition, except at the targets:

$$D_\pi(\sigma(s_t), a_t | \sigma(s^\odot)) \leftarrow \begin{cases} 1 + D_\pi(\sigma(s_{t+1}), a_{t+1} | \sigma(s^\odot)) & \text{if } s_{t+1} \neq s^\odot \\ 1 & \text{if } s_{t+1} = s^\odot \\ \infty & \text{if } s_{t+1} \text{ is terminal and } s_{t+1} \neq s^\odot \end{cases}$$

where $a_{t+1} \sim \pi(\cdot | \sigma(s_{t+1}), \sigma(s^\odot))$. The cumulative discount is then recovered by replacing the support of the output distance histogram with the corresponding discounts. Additionally, the learned distance is used to prune unwanted checkpoints to simplify the proxy problem, as well as prune far-fetched edges. The details of pruning will be presented shortly.

Please refer to the Appendix D.1 for the properties of the learning rules for $\hat{v}_\pi$ and $\hat{\gamma}_\pi$.

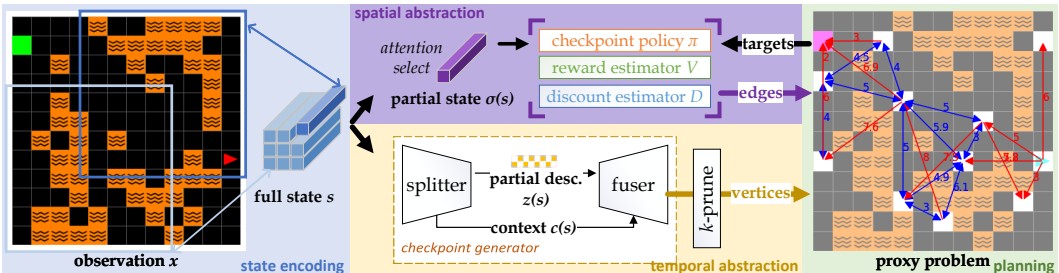

Figure 2: **Skipper Framework**: 1) Partial states consist of a few local fields, soft-selected via top-$k$ attention (Gupta et al., 2021). **Skipper**'s edge estimations and low-level behaviors $\pi$ are based on the partial states. 2) The checkpoint generator learns by splitting the full state into context and partial descriptions, and fusing them to reconstruct the input. It imagines checkpoints by sampling partial descriptions and combining them with the episodic contexts; 3) We prune the vertices and edges of the denser graphs to extract sparser proxy problems. Once a plan is formed, the immediate checkpoint target is used to condition the policy. In the proxy problem example, blue edges are estimated to be bidirectional and red edges have the other direction pruned.

### 3.4 PROBLEM 2: VERTEX GENERATION

The checkpoint generator aims to directly imagine the possible future states *without needing to know how exactly the agent might reach them nor worrying about if they are reachable*. The feasibility of checkpoint transitions will be abstracted by the connection estimates instead.

To make the checkpoint generator generalize well across diverse tasks, while still being able to capture the underlying causal mechanisms in the environment (a challenge for existing model-based methods) (Zhang et al., 2020), we propose that the checkpoint generator learns to split the state representation into two parts: an episodic context and a partial description. In a navigation problem, for example, as in Fig. 2, a context could be a representation of the map of a gridworld, and the partial description be the 2D-coordinates of the agent's location. In different contexts, the same partial description could correspond to very different states. Yet, within the same context, we should be able to recover the same state given the same partial description.

As shown in Fig. 2, this information split is achieved using two functions: the *splitter* $\mathcal{E}_{CZ}$, which maps the input state $S$ into a representation of a context $c(S)$ and a partial description $z(S)$, as well as the *fuser* $\bigoplus$ which, when applied to the input $\langle c, z \rangle$, recovers $S$. In order to achieve consistent context extraction across states in the same episode, at training time, we force the context to be extracted from other states in the same episode, instead of the input.

We sample in hindsight a diverse distribution of target encoded (full) states $S^{\odot}$, given any current $S_t$. Hence, we make the generator a conditional Variational AutoEncoder (VAE) (Sohn et al., 2015) which learns a distribution $p(S^{\odot}|C(S_t)) = \sum_z p(S^{\odot}|C(S_t), z)p(z|C(S_t))$, where $C(S_t)$ is the extracted context from $S_t$ and $z$s are the partial descriptions. We train the generator by minimizing the evidence lower bound on $\langle S_t, S^{\odot} \rangle$ pairs chosen with HER.

Similarly to Hafner et al. (2023), we constrain the partial description as a bundle of binary variables and train them with the straight-through gradients (Bengio et al., 2013). These binary latents can be easily sampled or composed for generation. Compared to models such as that in Director (Hafner et al., 2022), which generates intermediate goals given the on-policy trajectory, ours can generate and handle a more diverse distribution of states, beneficial for planning in novel scenarios.

#### 3.4.1 PRUNING

In this paper,we limit ourselves only to checkpoints from a return-unaware conditional generation model, leaving the question of how to improve the quality of the generated checkpoints for future work. Without learning, the proxy problem can be improved by making it more sparse, and making the proxy problem vertices more evenly spread in state space. To achieve this, we propose a pruning algorithm based on $k$-medoids clustering (Kaufman & Rousseeuw, 1990), which only requires pairwise distance estimates between states. During proxy problem construction, we first sample a larger number of checkpoints, and then cluster them and select the centers (which are always real states instead of imaginary weighted sums of state representations).

Notably, for sparse reward tasks, the generator cannot guarantee the presence of the rewarding checkpoints in the proposed proxy problem. We could remedy this by explicitly learning the generation of the rewarding states with another conditional generator. These rewarding states should be kept as vertices (immune from pruning).

In addition to pruning the vertices, we also prune the edges according to a distance threshold, *i.e.*, all edges with estimated distance over the threshold are deleted from the complete graph of the pruned vertices. This biases potential plans towards shorter-length, smaller-scale sub-problems, as far-away checkpoints are difficult for $\pi$ to achieve, trading optimality for robustness.

### 3.4.2 SAFETY & DELUSION CONTROL

Model-based HRL agents can be prone to blindly optimizing for objectives without understanding the consequences (Langosco et al., 2022; Paolo et al., 2022). We propose a technique to suppress delusions by exposing edge estimation to potentially delusional targets that do not exist in the experience replay buffer. Details and examples are provided in the Appendix.

## 4 RELATED WORKS & DISCUSSIONS

**Temporal Abstraction.** Resembling attention in human consciousness, choosing a checkpoint target is a selection towards certain decision points in the dimension of time, *i.e.* a form of temporal abstraction. Constraining options, **Skipper** learns the options targeting certain "outcomes", which dodges the difficulties of option collapse (Bacon et al., 2017) and option outcome modelling by design. The constraints indeed shift the difficulties to generator learning (Silver & Ciosek, 2012; Tang & Salakhutdinov, 2019). We expect this to entail benefits where states are easy to learn and generate, and / or in stochastic environments where the outcomes of unconstrained options are difficult to learn. Constraining options was also investigated in Sharma et al. (2019) in an unsupervised setting.

**Spatial Abstraction** is different from "state abstraction" (Sacerdoti, 1974; Knoblock, 1994), which evolved to be a more general concept that embraces mainly the aspect of state aggregation, *i.e.* state space partitioning (Li et al., 2006). Spatial abstraction, defined to capture the behavior of attention in conscious planning in the spatial dimensions, focuses on the **within-state** partial selection of the environmental state for decision-making. It corresponds naturally to the intuition that state representations should contain useful aspects of the environment, while not all aspects are useful for a particular intent. Efforts toward spatial abstraction are traceable to early hand-coded proof-of-concepts proposed in *e.g.* Dieterich (2000). Until only recently, attention mechanisms had primarily been used to construct state representations in model-free agents for sample efficiency purposes, without the focus on generalization (Mott et al., 2019; Manchin et al., 2019; Tang et al., 2020). In Fu et al. (2021); Zadaianchuk et al. (2020); Shah et al. (2021), 3 more recent model-based approaches, spatial abstractions are attempted to remove visual distractors. Concurrently, emphasizing on generalization, our previous work (Zhao et al., 2021) used spatially-abstract partial states in decision-time planning. We proposed an attention bottleneck to dynamically select a subset of environmental entities during the atomic-step forward simulation, without explicit goals provided as in Zadaianchuk et al. (2020). **Skipper**'s checkpoint transition is a step-up from our old approach, where we now show that spatial abstraction, an overlooked missing flavor, is as crucial for longer-term planning as temporal abstraction (Konidaris & Barto, 2009).

**Task Abstraction via Goal Composition** The early work McGovern & Barto (2001) suggested to use bottleneck states as subgoals to abstract given tasks into manageable steps. Nair et al. (2018); Florensa et al. (2018) use generative model to imagine subgoals while Eysenbach et al. (2019) search directly on the experience replay. In Kim et al. (2021), promising states to explore are generated and selected with shortest-path algorithms. Similar ideas have been attempted for guided exploration (Erraqabi et al., 2021; Kulkarni et al., 2016). Similar to Hafner et al. (2022), Czechowski et al. (2021) generate fixed-steps ahead subgoals for reasoning tasks, while Bagaria et al. (2021) augments the search graph by states reached fixed-steps ahead. Nasiriany et al. (2019); Xie et al. (2020); Shi et al. (2022) employ CEM to plan a chain of subgoals towards the task goal (Rubinstein, 1997b). **Skipper** utilizes proxy problems to abstract the given tasks via spatio-temporal abstractions (Bagaria et al., 2021). Checkpoints can be seen as sub-goals that generalize the notion of "landmarks" or "waypoints" in Sutton et al. (1999); Dieterich (2000); Savinov et al. (2018). Zhang et al. (2021) used latent landmark graphs as high-level guidance, where the landmarks are sparsified with weighted

sums in the latent space to compose subgoals. In comparison, our checkpoint pruning selects a subset of generated states, which is less prone to issues created by weighted sums.

**Planning Estimates.** Zhang et al. (2021) propose a distance estimate with an explicit regression. With TDMs (Pong et al., 2018), LEAP (Nasiriany et al., 2019) embraces a sparse intrinsic reward based on distances to the goal. Contrasting with our distance estimates, there is no empirical evidence of TDMs' compatibility with stochasticity and terminal states. Notably, Eysenbach et al. (2019) employs a similar distance learning scheme to learn the shortest path distance between states found in the experience replay; while our estimators learn the distance conditioned on evolving policies. Such aspect was also investigated in Nachum et al. (2018).

**Decision-Time HP Methods.** Besides LEAP (Nasiriany et al., 2019), decision-time planning with evolutionary algorithms was investigated in Nair & Finn (2020); Hafner et al. (2019).

## 5 EXPERIMENTS

As introduced in Sec. 2, our first goal is to test the zero-shot generalization ability of trained agents. To fully understand the results, it is necessary to have precise control of the difficulty of the training and evaluation tasks. Also, to validate if the empirical performance of our agents matches the formal analyses (Thm. 1), we need to know how close to the (optimal) ground truth our edge estimations and checkpoint policies are. These goals lead to the need for environments whose ground truth information (optimal policies, true distances between checkpoints, etc) can be computed. Thus, we base our experimental setting on the MiniGrid-BabyAI framework (Chevalier-Boisvert et al., 2018b;a; Hui et al., 2020). Specifically, we build on the experiments used in our previous works (Zhao et al., 2021; Alver & Precup, 2022): the agent needs to navigate to the goal from its initial state in gridworlds filled with terminal lava traps generated randomly according to a difficulty parameter, which controls their density. During evaluation, the agent is always spawned at the opposite side from the goals. During training, the agent's position is uniformly initialized to speed up training. We provide results for non-uniform training initialization in the Appendix.

These fully observable tasks prioritize on the challenge of reasoning over causal mechanisms over learning representations from complicated observations, which is not the focus of this work. Across all experiments, we sample training tasks from an environment distribution of difficulty $0.4$: each cell in the field has probability $0.4$ to be filled with lava while guaranteeing a path from the initial position to the goal. The evaluation tasks are sampled from a gradient of OOD difficulties - $0.25$, $0.35$, $0.45$ and $0.55$, where the training difficulty acts as a median. To step up the long(er) term generalization difficulty compared to existing work, we conduct experiments done on large, $12 \times 12$ maze sizes, (see the visualization in Fig 2). The agents are trained for $1.5 \times 10^6$ interactions.

We compare **Skipper** against two state-of-the-art Hierarchical Planning (HP) methods: LEAP (Nasiriany et al., 2019) and Director (Hafner et al., 2022). The comparative results include:

**Skipper-once**: A **Skipper** agent that generates one proxy problem at the start of the episode, and the replanning (choosing a checkpoint target based on the existing proxy problem) only triggers a quick re-selection of the immediate checkpoint target;

**Skipper-regen**: A **Skipper** agent that re-generates a proxy problem when replanning is triggered;

**modelfree**: A model-free baseline agent sharing the same base architecture with the **Skipper** variants - a prioritized distributional Double DQN (Dabney et al., 2018; Van Hasselt et al., 2016);

**Director**: A tuned Director agent (Hafner et al., 2022) fed with simplified visual inputs. Since Director discards trajectories that are not long enough for training purposes, we make sure that the same amount of training data is gathered as for the other agents;

**LEAP**: A re-implemented LEAP for discrete action spaces. Due to low performance, we replaced the VAE and the distance learning mechanisms with our counterparts. We waived the interaction costs for its generator pretraining stage, only showing the second stage of RL pretraining.

Please refer to the Appendix for more details and insights on these agents.

## 5.1 GENERALIZATION PERFORMANCE

Fig. 3 shows how the agents' generalization performance evolves during training. These results are obtained with 50 fixed sampled training tasks (different 50s for each seed), a representative configuration of different numbers of training tasks including $\{1, 5, 25, 50, 100, \infty\}^2$, whose results are in the Appendix. In Fig. 3 a), we observe how well an agent performs on its training tasks. If an agent performs well here but badly in b), c), d) and e), *e.g.* the **modelfree** baseline, then we suspect that it overfitted on training tasks, likely indicating a reliance on memorization (Cobbe et al., 2020).

We observe a (statistically-)significant advantage in the generalization performance of the **Skipper** agents throughout training. We have also included significance tests and power analyses (Colas et al., 2018; Patterson et al., 2023) in the Appendix, together with results for other training configurations. The **regen** variant exhibits dominating performance over all others. This is likely due to the frequent reconstruction of the graph makes the agent less prone to being trapped in a low-quality proxy problem and provides extra adaptability in novel scenarios (more discussions in the Appendix). During training, **Skipper**s behave less optimally than expected, despite the strong generalization on evaluation tasks. As our ablation results and theoretical analyses consistently show, such a phenomenon is a composite outcome of inaccuracies both in the proxy problem and the checkpoint policy. One major symptom of an inaccurate proxy problem is that the agent would chase delusional targets. We address this behavior with the delusion suppression technique, to be discussed in the Appendix.

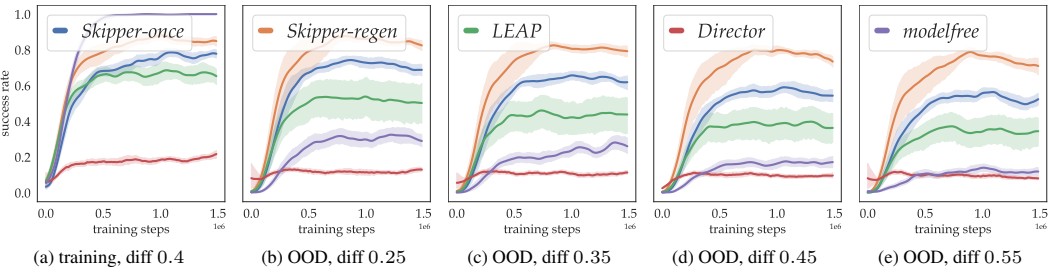

(a) training, diff 0.4    (b) OOD, diff 0.25    (c) OOD, diff 0.35    (d) OOD, diff 0.45    (e) OOD, diff 0.55

Figure 3: **Generalization Performance of Agents During Training**: the $x$-axes correspond to training progress, while the aligned $y$-axes represent the success rate of episodes (optimal is 1.0). Each agent is trained with 50 tasks. Each data point is the average success rate over 20 evaluation episodes, and each error bar (95% confidence interval) is processed from 20 independent seed runs. Training tasks performance is shown in (a) while OOD evaluation performance is shown in (b), (c), (d), (e).

Better than the **modelfree** baseline, LEAP obtains reasonable generalization performance, despite the extra budget it needs for pretraining. In the Appendix, we show that LEAP benefits largely from the delusion suppression technique. This indicates that optimizing for a path in the latent space may be prone to errors caused by delusional subgoals. Lastly, we see that the Director agents suffer in these experiments despite their good performance in the single environment experimental settings reported by Hafner et al. (2022). We present additional experiments in the Appendix to show that Director is ill-suited for generalization-focused settings: Director still performs well in single environment configurations, but its performance deteriorates fast with more training tasks. This indicates poor scalability in terms of generalization, a limitation to its application in real-world scenarios.

## 5.2 SCALABILITY OF GENERALIZATION PERFORMANCE

Like Cobbe et al. (2020), we investigate the scalability of the agents' generalization abilities across different numbers of training tasks. To this end, in Fig. 4, we present the results of the agents' final evaluation performance after training over different numbers of training tasks.

With more training tasks, **Skipper**s and the baseline show consistent improvements in generalization performance. While both LEAP and Director behave similarly as in the previous subsection, notably, the **modelfree** baseline can reach similar performance as **Skipper**, but only when trained on a different task in each episode, which is generally infeasible in the real world beyond simulation.

---

$^2\infty$ training tasks mean that an agent is trained on a different task for each episode. In reality, this may lead to prohibitive costs in creating the training environment.

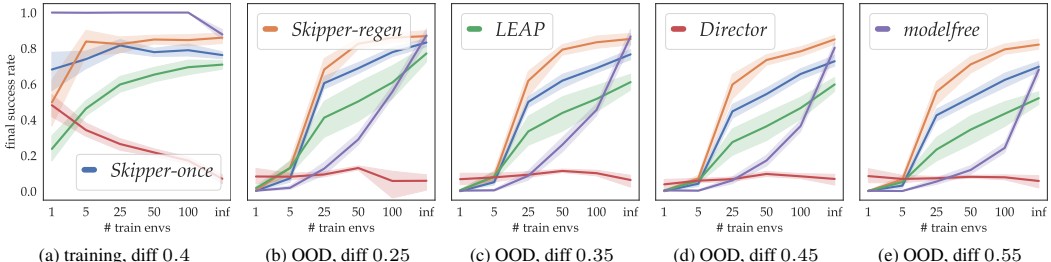

(a) training, diff 0.4    (b) OOD, diff 0.25    (c) OOD, diff 0.35    (d) OOD, diff 0.45    (e) OOD, diff 0.55

Figure 4: **Generalization Performance of Agents on Different Numbers of Training Tasks**: each data point and corresponding error bar (95% confidence interval) are based on the final performance from 20 independent seed runs. Training task performance is shown in (a) while OOD performance is shown in (b), (c), (d), (e). Notably, the **Skipper** agents as well as the adapted LEAP behave poorly during training when being trained on only one task, as the split of context and partial information cannot be achieved. Training on one task invalidates the purpose of the proposed generalization-focused checkpoint generator.

## 5.3 ABLATION & SENSITIVITY STUDIES

In the Appendix, we present ablation results confirming the effectiveness of delusion suppression, $k$-medoids pruning and the effectiveness of spatial abstraction via the local perception field. We also provide sensitivity study for the number of checkpoints in each proxy problem.

## 5.4 SUMMARY OF EXPERIMENTS

Within the scope of the experiments, we conclude that **Skipper** provides benefits for generalization; And it achieves better generalization when exposed to more training tasks;

From the content presented in the Appendix, we deduce additionally that:

- Spatial abstraction based on the local perception field is crucial for the scalability of the agents;

- **Skipper** performs well by reliably decomposing the given tasks, and achieving the sub-tasks robustly. Its performance is bottlenecked by the accuracy of the estimated proxy problems as well as the checkpoint policies, which correspond to goal generalization and capability generalization, respectively, identified in Langosco et al. (2022). This matches well with our theory. The proposed delusion suppression technique (in Appendix) is effective in suppressing plans with non-existent checkpoints as targets, thereby increasing the accuracy of the proxy problems;

- LEAP fails to generalize well within its original form and can generalize better when combined with the ideas proposed in this paper; Director may generalize better only in domains where long and informative trajectory collection is possible;

- We verified empirically that, as expected, **Skipper** is compatible with stochasticity.

## 6 CONCLUSION & FUTURE WORK

Building on previous work on spatial abstraction (Zhao et al., 2021), we proposed, analyzed and validated **Skipper**, which generalizes its learned skills better than the compared methods, due to its combined spatio-temporal abstractions. A major unsatisfactory aspect of this work is that we generated checkpoints at random by sampling the partial description space. Despite the pruning mechanisms, the generated checkpoints, and thus temporal abstraction, do not prioritize the predictable, important states that matter to form a meaningful plan (Şimşek & Barto, 2004). We would like to continue investigating the possibilities along this line. Additionally, we would like to explore other environments where the accuracy assumption (in Thm. 1) can meaningfully hold, *i.e.* beyond sparse reward cases.

## 7 REPRODUCIBILITY STATEMENT

The results presented in the experiments are **fully**-reproducible with the open-sourced repository `https://github.com/mila-iqia/Skipper`.

## 8 ACKNOWLEDGEMENTS

Mingde is grateful for the financial support of the FRQ (Fonds de recherche du Québec) and the collaborative efforts during his time at Microsoft Research Montreal. We want to thank Riashat Islam, Ziyu Wang for the discussions in the initial stages of the project. We appreciate the computational resources allocated to us from Mila and McGill university.

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

## A  APPENDIX

Please use the following to quickly navigate to your points of interest.

- **Weaknesses & Limitations** (Sec. B)

- **Skipper Algorithmic Details** (Sec. C): pseudocodes, $k$-medoids based pruning, delusion suppression

- **Theoretical Analyses** (Sec. D.1): detailed proofs, discussions

- **Implementation Details** (Sec. E): for **Skipper**, **LEAP** and **Director**

- **More Experiments** (Sec. F): experimental results that cannot be presented in the main paper due to page limit

- **Ablation Tests & Sensitivity Analyses** (Sec. G)

## B  WEAKNESSES & LIMITATIONS

We would like to expand the discussions on the limitations to the current form of **Skipper**, as well as the design choices that we seek to improve in the future:

- We generate future checkpoints at random by sampling the partial description space. Despite the post-processing such as pruning, the generated checkpoints do not prioritize on the predictable, important states that matter the most to form a meaningful long-term plan.

- The current implementation is intended for pixel input fully-observable tasks with discrete state and action spaces. Such a minimalistic form is because we wish to isolate the unwanted challenges from other factors that are not closely related to the idea of this work, as well as to make the agent as generalist as possible. **Skipper** is naturally compatible with continuous actions spaces and the only thing we will need to do is to replace the baseline agent with a compatible one such as TD3 (Fujimoto et al., 2018); on the other hand, for continuous state spaces, the identification of the achievement of a checkpoint becomes tricky. This is due to the fact that a strict identity between the current state and the target checkpoint may be ever established, we either must adopt a distance measure for approximate state equivalence, or rely on the equivalence of the partial descriptions (which is adopted in the current implementation). We intentionally designed the partial descriptions to be in the form of bundles of binary variables, so that this comparison could be done fast and trivially for any forms of the state space; for partial observability, despite that no recurrent mechanism has been incorporated in the current implementation, the framework is not incompatible. To implement that, we will need to augment the state encoder with recurrent or memory mechanisms and we need to make the checkpoint generator directly work over the learned state representations. We acknowledge that future work is needed to verify **Skipper**'s performance on the popular partially-observable benchmark suites, which requires the incorporation of components to handle partial observability as well as scaling up the architectures for more expressive power;

- We do not know the precise boundaries of the motivating theory on proxy problems, since it only indicates performance guarantees on the condition of estimation accuracy, which in turn does not correspond trivially to a set of well-defined problems. We are eager to explore, outside the scope of sparse-reward navigation, how this approach can be used to facilitate better generalization, and at the same time, try to find more powerful theories that guide us better;

## C  **Skipper**'S ALGORITHMIC DETAILS

### C.1  OVERALL **Skipper** FRAMEWORK (PSEUDO-CODE)

The pseudocode of **Skipper** is provided in Alg. 1, together with the hyperparameters used in our implementation.

---

**Algorithm 1: Skipper** with Random Checkpoints (implementation choice in purple)

---

**for** *each episode* **do**

    // — start of the **subroutine** to construct the proxy problem

    generate more than necessary (32) checkpoints by sampling from the partial descriptions
      given the extracted context from the initial state;

    $(k = 12)$-medoid pruning upon estimated distances among all checkpoints; // prune vertices

    use estimators to annotate the edges between the nodes (including a terminal state estimator
      to correct the estimates);

    prune edges that are too far-fetched according to distance estimations (threshold set to be 8,
      same as replan interval); // prune edges

    // — end of the **subroutine** to construct the proxy problem

    **for** *each agent-environment interaction step until termination of episode* **do**

        **if** *decided to explore (DQN-style annealing $\epsilon$-greedy)* **then**

            take a random action;

        **else**

            **if** *abstract problem just constructed **or** a checkpoint / timeout reached ($\geq 8$ steps
              since last planned)* **then**

                [**OPTIONAL RE-GENERATION**] call the **subroutine** above for
                  **Skipper-regen**;

                run value iteration (for 5 iterations) on the proxy problem, select the target
                  checkpoint;

            follow the action suggested by the checkpoint-achieving policy;

        **if** *time to train (every 4 actions)* **then**

            sample hindsight transitions and train checkpoint-achieving policy, estimators
              (including a teriminal state estimator) and checkpoint generator;

            [**OPTIONAL DELUSION CONTROL**]: train estimators using generated
              checkpoints;

        save interaction into the trajectory experience replay;

    convert trajectory into HER samples (relabel 4 random states as additional goals);

---

## C.2   $k$-MEDOIDS BASED PRUNING

We present the pseudocode of the modified $k$-medoids algorithm for pruning overcrowded checkpoints in Alg. 2. Note that the presented pseudocode is optimized for readers' understanding, while the actual implementation is parallelized. The changes upon the original $k$-medoids algorithm is marked in purple, which implement a forced preservation of data points: when $k$-medoids is called after the unpruned graph is constructed, $\mathcal{S}_\vee$ is set to be the set containing the goal state only. This is intended to span more uniformly in the state space with checkpoints, while preserving the goal.

Let the estimated distance matrix be $D$, where each element $d_{i}j$ represents the estimated trajectory length it takes for $\pi$ to fulfill the transition from checkpoint $i$ to checkpoint $j$. Since $k$-medoids cannot handle infinite distances (*e.g.* from a terminal state to another state), the distance matrix $D$ is truncated, and then we take the elementwise minimum between the truncated $D$ and $D^T$ to preserve the one-way distances. The matrix containing the elementwise minimums would be the input of the pruning algorithm.

---

**Algorithm 2:** Checkpoint Pruning with $k$-medoids

---

**Data:** $X = \{x_1, x_2, \ldots, x_n\}$ (state indices), $D$ (estimated distance matrix), $\mathcal{S}_\vee$ (states that must be kept), $k$ (#checkpoints to keep)

**Result:** $\mathcal{S}_\odot \equiv \{M_1, M_2, \ldots, M_k\}$ (checkpoints kept)

---

Initialize $\mathcal{S}_\odot \equiv \{M_1, M_2, \ldots, M_k\}$ randomly from $X$
make sure $\mathcal{S}_\vee \subset \mathcal{S}_\odot$
**repeat**
    Assign each data point $x_i$ to the nearest medoid $M_j$, forming clusters $C_1, C_2, \ldots, C_k$;
    **foreach** *medoid $M_j$* **do**
        Calculate the cost $J_j$ of $M_j$ as the sum of distances between $M_j$ and the data points in $C_j$;
    Find the medoid $M_j$ with the lowest cost $J_j$;
    **if** *$M_j$ changes* **then**
        make sure $\mathcal{S}_\vee \subset \mathcal{S}_\odot$
        Replace $M_j$ with the data point in $C_j$ that minimizes the total cost;
**until** *Convergence (no cost improvement)*;

---

## C.3   DELUSION SUPPRESSION

RL agents are prone to blindly optimizing for an intrinsic objective without fully understanding the consequences of its actions. Particularly in model-based RL or in Hierarchical RL (HRL), there is a significant risk posed by the agents trying to achieve delusional future states that do not exist or beyond the safety constraints. With a use of a learned generative model, as in **Skipper** and other HP frameworks, such risk is almost inevitable, because of uncontrollable generalization effects.

Generalization abilities of the generative models are a double-edged sword. The agent would take advantage of its potentials to propose novel checkpoints to improve its behavior, but is also at risk of wanting to achieve non-existent unknown consequences. In **Skipper**, checkpoints imagined by the generative model could correspond to non-existent "states" that would lead to delusional edge estimates and therefore confuse planning. For instance, arbitrarily sampling partial descriptions may result in a delusional state where the agent is in a cell that can never be reached from the initial states. Since such states do not exist in the experience replay, the estimators will have not learned how to handle them appropriately when encountered in the generated proxy problem during decision time. We present a resulting failure mode in Fig. 5.

To address such concerns, we propose an optional auxiliary training procedure that makes the agent stay further away from delusional checkpoints. Due to the favorable properties of the update rules of $D_\pi$ (in fact, $V_\pi$ as well), all we have to do is to replace the hindsight-sampled target states with generated checkpoints, which contain non-existent states. Then, the auxiliary rewards will all converge to the minimum in terms of favorability on the non-existent states. This is implemented trivially by adding a loss to the original training loss for the distance estimator, which we give a $0.25$ scaling for stability.

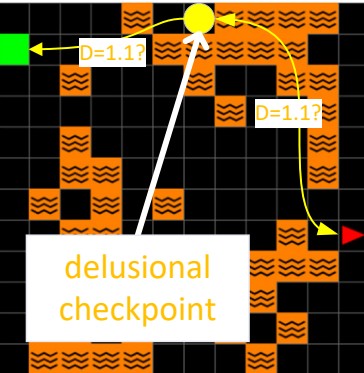

Figure 5: **Example of Failure Caused by Delusions**: we illustrate an instance of chasing delusional checkpoint in one of our experimental runs by **Skipper**. The distance (discount) estimator, probably due to the ill-generalization, estimates that the delusional checkpoint (yellow) is very close to every other state. A resulting plan was that the agent thought it could reach any far-away checkpoints by using the delusional state to form a shortcut: the goal that was at least 17 steps away would be reached in 2.2.

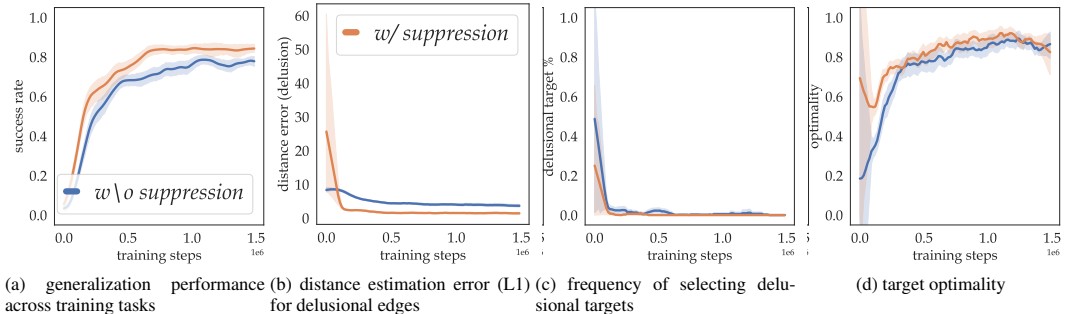

(a) generalization performance across training tasks  (b) distance estimation error (L1) for delusional edges  (c) frequency of selecting delusional targets  (d) target optimality

Figure 6: **Performance of Skipper-once with the proposed Delusion Suppression Technique**: each curve and corresponding error bar (95% CI) are processed from 20 independent seed runs. a) the performance across training tasks is shown. A more optimal performance can be achieved with **Skipper**-once in training tasks, when delusions are suppressed; b) During training interactions, the error in estimated (truncated) distance from and to delusional targets are significantly reduced with the technique; c) The frequency of selecting a delusional target is reduced to almost negligible during the whole training process; d) The optimality of target checkpoint during training can be improved by the suppression. Each agent is trained with 50 environments and each curve is processed from 20 independent seed runs.

---

**Algorithm 3:** Delusion Suppression

---

// This whole code block should be injected into the training loop if used

generate using the checkpoint generator, from the sampled batch of encoded states, the target states (to overwrite those relabelled in the HER) *i.e.* replace $\langle s_t, a_t, r_{t+1}, s_{t+1}, s^{\odot} \rangle$ with $\langle s_t, a_t, r_{t+1}, s_{t+1}, s^{\odot}_* \rangle$, where $s^{\odot}_*$ are generated from the context of $s_t$

train the distance estimator $D$ as if these are sampled from the HER

---

We provide analytic results and related discussion for **Skipper-once** agents trained with the proposed delusion suppression technique on 50 training tasks in Fig. 6. The delusion suppression technique is not enabled by default because it was not introduced in the main manuscript due to the page limits.

The delusion suppression technique can also be used to help us understand the failure modes of LEAP in Sec. E.2.4.

## D  THEORETICAL ANALYSES

### D.1  UPDATE RULES FOR EDGE ESTIMATION

First, we want to show that the update rules proposed in the main paper indeed estimate the desired cumulative discount and reward.

The low-level checkpoint-achieving policy $\pi$ is trained with an intrinsic reward to reach target state $s^{\odot}$. The cumulative reward and cumulative discount are estimated by applying policy evaluation given $\pi$, on the two sets of auxiliary reward signals, respectively.

For the cumulative discounted reward random variable:

$$V_\pi(s_t, a_t | s^{\odot}) = R(s_t, a_t, S_{t+1}) + \gamma V_\pi(S_{t+1}, A_{t+1} | s^{\odot}) \tag{5}$$

$$= \sum_{\tau=t}^{\infty} \gamma^{\tau-t} R(S_\tau, A_\tau, S_{\tau+1}), \tag{6}$$

where $S_{t+1} \sim p(\cdot|s_t, a_t)$, $A_{t+1} \sim \pi(\cdot|S_{t+1}, s^{\odot})$, and with $V_\pi(S_{t+1}, A_{t+1}|s^{\odot}) = 0$ if $S_{t+1} = s^{\odot}$. We overload the notation as follows: $V_\pi(s|s^{\odot}) \doteq V_\pi(s, A|s^{\odot})$ with $A \sim \pi(\cdot|s, s^{\odot})$.

The cumulative discount random variable denotes the event that the trajectory did not terminate before reaching the target $s^{\odot}$:

$$\Gamma_\pi(S_t, A_t | s^{\odot}) = \gamma \cdot \Gamma_\pi(S_{t+1}, A_{t+1} | s^{\odot}), \tag{7}$$

$$= \gamma^{T_\perp - t} \mathbb{I}\{S_{T_\perp} = s^{\odot}\}, \tag{8}$$

where $T_\perp$ denotes the timestep when the trajectory terminates, and with $\Gamma_\pi(S_{t+1}, A_{t+1}|s^{\odot}) = 1$ if $S_{t+1} = s^{\odot}$ and $\Gamma_\pi(S_{t+1}, A_{t+1}|s^{\odot}) = 0$ if $S_{t+1} \neq s^{\odot}$ is terminal. We overload the notation as follows: $\Gamma_\pi(s_t|s^{\odot}) \doteq \Gamma_\pi(s_t, A_t|s^{\odot})$ with $A_{t+1} \sim \pi(\cdot|S_{t+1}, s^{\odot})$.

Note that, for the sake of simplicity, we take here the view that the terminality of states is deterministic, but this is not reductive as any state with a stochastic terminality can be split into two identical states: one that is deterministically non-terminal and the other that is deterministically terminal. Note also that we could adopt the view that the discount factor is the constant probability of the trajectory to not terminate.

### D.2  PERFORMANCE BOUND

We are going to denote the expected cumulative discounted reward, *a.k.a.* the state-action value with $q_\pi \doteq \mathbb{E}_\pi[V]$, and let $\hat{q}_\pi$ be our estimate for it. We are also going to consider the state value $v_\pi(s|s^{\odot}) \doteq \sum_a \pi(a|s, s^{\odot}) q_\pi(s, a|s^{\odot})$ and its estimate $\hat{v}_\pi$. Similarly, we denote the expected cumulative discount with $\gamma_\pi \doteq \mathbb{E}_\pi[\Gamma]$ and its estimate with $\hat{\gamma}_\pi$.

We are in the presence of a hierarchical policy. The high level policy $\mu$ consists in (potentially) stochastically picking a sequence of checkpoints. The low-level policy is implemented by $\pi$ which is assumed to be given and fixed for the moment. The composite policy $\mu \circ \pi$ is non-Markovian: it depends both on the current state and the current checkpoint goal. So there is no notion of state value, except when we arrive at a checkpoint, *i.e.* when a high level action (checkpoint selection) needs to be chosen.

Proceeding further, we adopt the view where the discounts are a way to represent the hazard of the environment: $1 - \gamma$ is the probability of sudden trajectory termination. In this view, $v_\pi$ denotes the (undiscounted: there is no more discounting) expected sum of reward before reaching the next

checkpoint, and more interestingly $\gamma_\pi$ denotes the binomial random variable of non-termination during the transition to the selected checkpoint.

Making the following assumption that the trajectory terminates almost surely when reaching the goal, *i.e.* $\gamma_\pi(s_i, s_g) = 0, \forall s_i$, the gain $V$ can be written:

$$V_0 = V(S_0^{\odot}|S_1^{\odot}) + \Gamma(S_0^{\odot}|S_1^{\odot})V_1 = \sum_{k=0}^{\infty} V(S_k^{\odot}|S_{k+1}^{\odot}) \prod_{i=0}^{k-1} \Gamma(S_i^{\odot}|S_{i+1}^{\odot}), \qquad (9)$$

where $S_{k+1} \sim \mu(\cdot|S_k)$, where $V(S_k^{\odot}|S_{k+1}^{\odot})$ is the gain obtained during the path between $S_k^{\odot}$ and where $S_{k+1}^{\odot}$, and $\Gamma(S_k^{\odot}|S_{k+1}^{\odot})$ is either 0 or 1 depending whether the trajectory terminated or reached $S_{k+1}^{\odot}$. If we consider $\mu$ as a deterministic planning routine over the checkpoints, then the action space of $\mu$ boils down to a list of checkpoints $\{s_0^{\odot} = s_0, s_1^{\odot}, \cdots, s_n^{\odot} = s_g\}$. Thanks to the Markovian property in checkpoints, we have independence between $V_\pi$ and $\Gamma_\pi$, therefore for the expected value of $\mu \circ \pi$, we have:

$$v_{\mu \circ \pi}(s_0) \doteq \mathbb{E}_{\mu \circ \pi}[V|S_0 = s_0] = \sum_{k=0}^{\infty} v_\pi(s_k^{\odot}|s_{k+1}^{\odot}) \prod_{i=0}^{k-1} \gamma_\pi(s_i^{\odot}|s_{i+1}^{\odot}) \qquad (10)$$

Having obtained the ground truth value, in the following, we are going to consider the estimates which may have small error terms:

$$|v_\pi(s) - \hat{v}_\pi(s)| < \epsilon_v v_{\max} \ll (1-\gamma)v_{\max} \qquad \text{and} \qquad |\gamma_\pi(s) - \hat{\gamma}_\pi(s)| < \epsilon_\gamma \ll (1-\gamma)^2 \qquad \forall s. \tag{11}$$

We are looking for a performance bound, and assume without loss of generality that the reward function is non-negative, *s.t.* the values are guaranteed to be non-negative as well. We provide an upper bound:

$$\hat{v}_{\mu \circ \pi}(s) \doteq \sum_{k=0}^{\infty} \hat{v}_\pi(s_k^{\odot}|s_{k+1}^{\odot}) \prod_{i=0}^{k-1} \hat{\gamma}_\pi(s_i^{\odot}|s_{i+1}^{\odot}) \tag{12}$$

$$\leq \sum_{k=0}^{\infty} \left(v_\pi(s_k^{\odot}|s_{k+1}^{\odot}) + \epsilon_v v_{\max}\right) \prod_{i=0}^{k-1} \left(\gamma_\pi(s_i^{\odot}|s_{i+1}^{\odot}) + \epsilon_\gamma\right) \tag{13}$$

$$\leq v_{\mu \circ \pi}(s) + \sum_{k=0}^{\infty} \epsilon_v v_{\max} \prod_{i=0}^{k-1} \left(\gamma_\pi(s_i^{\odot}|s_{i+1}^{\odot}) + \epsilon_\gamma\right) + \sum_{k=0}^{\infty} \left(v_\pi(s_k^{\odot}|s_{k+1}^{\odot}) + \epsilon_v v_{\max}\right) k \epsilon_\gamma \gamma^k + o(\epsilon_v + \epsilon_\gamma) \tag{14}$$

$$\leq v_{\mu \circ \pi}(s) + \epsilon_v v_{\max} \sum_{k=0}^{\infty} \gamma^k + \epsilon_\gamma v_{\max} \sum_{k=0}^{\infty} k \gamma^k + o(\epsilon_v + \epsilon_\gamma) \tag{15}$$

$$\leq v_{\mu \circ \pi}(s) + \frac{\epsilon_v v_{\max}}{1-\gamma} + \frac{\epsilon_\gamma v_{\max}}{(1-\gamma)^2} + o(\epsilon_v + \epsilon_\gamma) \tag{16}$$

Similarly, we can derive a lower bound:

$$\hat{v}_{\mu \circ \pi}(s) \doteq \sum_{k=0}^{\infty} \hat{v}_{\pi}(s_k^{\odot}|s_{k+1}^{\odot}) \prod_{i=0}^{k-1} \hat{\gamma}_{\pi}(s_i^{\odot}|s_{i+1}^{\odot}) \tag{17}$$

$$\geq \sum_{k=0}^{\infty} \left( v_{\pi}(s_k^{\odot}|s_{k+1}^{\odot}) - \epsilon_v v_{\max} \right) \prod_{i=0}^{k-1} \left( \gamma_{\pi}(s_i^{\odot}|s_{i+1}^{\odot}) - \epsilon_{\gamma} \right) \tag{18}$$

$$\geq v_{\mu \circ \pi}(s) - \sum_{k=0}^{\infty} \epsilon_v v_{\max} \prod_{i=0}^{k-1} \left( \gamma_{\pi}(s_i^{\odot}|s_{i+1}^{\odot}) - \epsilon_{\gamma} \right) - \sum_{k=0}^{\infty} \left( v_{\pi}(s_k^{\odot}|s_{k+1}^{\odot}) - \epsilon_v v_{\max} \right) k \epsilon_{\gamma} \gamma^k + o(\epsilon_v + \epsilon_{\gamma}) \tag{19}$$

$$\geq v_{\mu \circ \pi}(s) - \epsilon_v v_{\max} \sum_{k=0}^{\infty} \gamma^k - \epsilon_{\gamma} v_{\max} \sum_{k=0}^{\infty} k \gamma^k + o(\epsilon_v + \epsilon_{\gamma}) \tag{20}$$

$$\geq v_{\mu \circ \pi}(s) - \frac{\epsilon_v v_{\max}}{1 - \gamma} - \frac{\epsilon_{\gamma} v_{\max}}{(1 - \gamma)^2} + o(\epsilon_v + \epsilon_{\gamma}) \tag{21}$$

We may therefore conclude that $\hat{v}_{\mu \circ \pi}$ equals $v_{\mu \circ \pi}$ up to an accuracy of $\frac{\epsilon_v v_{\max}}{1-\gamma} + \frac{\epsilon_{\gamma} v_{\max}}{(1-\gamma)^2} + o(\epsilon_v + \epsilon_{\gamma})$. Note that the requirement for the reward function to be positive is only a cheap technical trick to ensure we bound in the right direction of $\epsilon_{\gamma}$ errors in the discounting, but that the theorem would still stand if it were not the case.

### D.3 NO ASSUMPTION ON OPTIMALITY

If the low-level policy $\pi$ is perfect, then the best high-level policy $\mu$ is to choose directly the goal as target[3]. Our approach assumes that it would be difficult to learn effectively a $\pi$ when the target is too far, and that we would rather use a proxy to construct a path with shorter-distance transitions. Therefore, we'll never want to make any optimality assumption on $\pi$, otherwise our approach is pointless. These theories we have initiated makes no assumption on $\pi$.

The Theorem provides guarantees on the solution to the overall problem. The quality of the solution depends on both the quality of the estimates (distances/discounts, rewards) and the quality of the policy, as the theorem guarantees accuracy to the solution of the overall problem given a current policy, which should evolve towards optimal during training. This means bad policy with good estimation will lead to an accurate yet bad overall solution. No matter the quality of the policy, with a bad estimation, it will result in a poor estimate of solutions. Only a near-optimal policy and good estimation will lead to a near-optimal solution.

## E IMPLEMENTATION DETAILS FOR EXPERIMENTS

### E.1 Skipper

#### E.1.1 TRAINING

The agent is based on a distributional prioritized double DQN. All the trainable parameters are optimized with Adam at a rate of $2.5 \times 10^{-4}$ (Kingma & Ba, 2014), with a gradient clipping by value (maximum absolute value 1.0). The priorities for experience replay sampling are equal to the per-sample training loss.

#### E.1.2 FULL STATE ENCODER

The full-state encoder is a two layered residual block (with kernel size 3 and doubled intermediate channels) combined with the 16-dimensional bag-of-words embedder of BabyAI (Hui et al., 2020).

---

[3]A triangular inequality can be shown that with a perfect $\pi$ and a perfect estimate of $v_{\pi}$ and $\gamma_{\pi}$, the performance will always be minimized by selecting $s_1^{\odot} = s_g$.

### E.1.3 PARTIAL STATE SELECTOR (SPATIAL ABSTRACTION)

The selector $\sigma$ is implemented with one-head (not multiheaded, therefore the output linear transformation of the default multihead attention implementation in PyTorch is disabled.) top-4 attention, with each local perceptive field of size $8 \times 8$ cells. Layer normalization (Ba et al., 2016) is used before and after the spatial abstraction.

### E.1.4 ESTIMATORS

The estimators, which operate on the partial states, are 3-layered MLPs with 256 hidden units.

An additional estimator for termination is learned, which instead of taking a pair of partial states as input, takes only one, and is learned to classify terminal states with cross-entropy loss. The estimated distance from terminal states to other states would be overwritten with $\infty$. The internal $\gamma$ for intrinsic reward of $\pi$ is 0.95, while the task $\gamma$ is 0.99

The estimators use C51 distributional TD learning (Dabney et al., 2018). That is, the estimators output histograms (softmax over vector outputs) instead of scalars. We regress the histogram towards the targets, where these targets are skewed histograms of scalar values, towards which KL-divergence is used to train. At the output, there are 16 bins for each histogram estimation (value for policy, reward, distance).

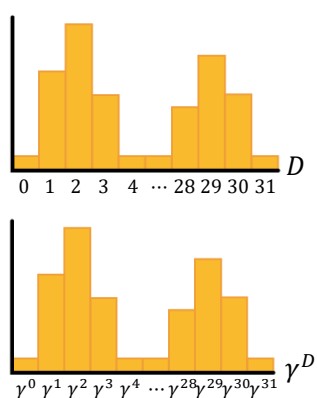

Figure 7: **Estimating Distributions of Discount and Distance with the Same Histogram**: by transplanting the support with the corresponding discount values, the distribution of the cumulative discount can be inferred.

### E.1.5 RECOVERING DISCOUNTS FROM DISTANCES

We recover the distribution of the cumulative discount by replacing the support of the discretized truncated distances with the corresponding discounts, as illustrated in Fig. 7. This addresses the problem of $\mathbb{E}[\gamma^D] \neq \gamma^{\mathbb{E}[D]}$, as the probability of having a trajectory length of 4 under policy $\pi$ from state $s_t$ to $s_\odot$ is the same as a trajectory having discount $\gamma^4$.

### E.1.6 CHECKPOINT GENERATOR

Despite **Skipper** is designed to have the generator work on state level, that is, it should take learned state representations as inputs and have state representations as outputs, in our experiments, the generator actually operates on observation inputs and outputs. This is because of the preferred compactness of the observations and the equivalence to full states under full observability in our experiments.

The context extractor $\mathcal{E}_c$ is a 32-dimensional BabyAI BOW embedder. It encodes an input observation into a representation of the episodic context.

The partial description extractor $\mathcal{E}_z$ is made of a 32-dimensional BabyAI BOW embedder, followed by 3 aforementioned residual blocks with $3 \times 3$ convolutions (doubling the feature dimension every time) in between, ended by global maxpool and a final linear projection to the latent weights. The partial descriptions are bundles of 6 binary latents, which could represent at most 64 "kinds" of checkpoints. Inspired by VQ-VAE (Van Den Oord et al., 2017), we use the argmax of the latent weights as partial descriptions, instead of sampling according to the softmax-ed weights. This enables easy comparison of current state to the checkpoints in the partial description space, because each state deterministically corresponds to one partial description. We identify reaching a target checkpoint if the partial description of the current state matches that of the target.

The fusing function first projects linearly the partial descriptions to a 128-dimensional space and then uses deconvolution to recover an output which shares the same size as the encoded context. Finally, a residual block is used, followed by a final $1x1$ convolution that downscales the concatenation of context together with the deconv'ed partial description into a 2D weight map. The agent's location is taken to be the argmax of this weight map.

The whole checkpoint generator is trained end-to-end with a standard VAE loss. That is the sum of a KL-divergence for the agent's location, and the entropy of partial descriptions, weighted by $2.5 \times 10^{-4}$, as suggested in `https://github.com/AntixK/PyTorch-VAE`. Note that the per-sample losses in the batches are not weighted for training according to priority from the experience replay.

We want to mention that if one does not want to generate non-goal terminal states as checkpoints, we could also seek to train on reversed $\langle S^{\odot}, S_t \rangle$ pairs. In this case, the checkpoints to reconstruct will never be terminal.

### E.1.7 HER

Each experienced transition is further duplicated into $4$ hindsight transitions at the end of each episode. Each of these transitions is combined with a randomly sampled observation from the same trajectory as the relabelled "goal". The size of the hindsight buffer is extended to $4$ times that of the baseline that does not learn from hindsight accordingly, that is, $4 \times 10^6$.

### E.1.8 PLANNING

As introduced, we use value iteration over options (Sutton et al., 1999) to plan over the proxy problem represented as an SMDP. We use the matrix form $Q = R_{S \times S} + \Gamma V$, where $R$ and $\Gamma$ are the estimated edge matrices for cumulative rewards, respectively. Note that this notation is different from the ones we used in the manuscript. The checkpoint value $V$, initialized as all-zero, is taken on the maximum of $Q$ along the checkpoint target (the actions for $\mu$) dimension. When planning is initiated during decision time, the value iteration step is called $5$ times. We do not run until convergence since with low-quality estimates during the early stages of the learning, this would be a waste of time. The edges from the current state towards other states are always set to be one-directional, and the self-loops are also removed. This means the first column as well as the diagonal elements of $R$ and $\Gamma$ are all zeros. Besides pruning edges based on the distance threshold, as introduced in the main paper, the terminal estimator is also used to prune the matrices $R$ and $\Gamma$: the rows corresponding to the terminal states are all zeros.

The only difference between the two variants, *i.e.* **Skipper-once** and **Skipper-regen** is that the latter variant would discard the previously constructed proxy problem and construct a new one every time the planning is triggered. This introduces more computational effort while lowering the chance that the agent gets "trapped" in a bad proxy problem that cannot form effective plans to achieve the goal. If such a situation occurs with **Skipper-regen**, as long as the agent does not terminate the episode prematurely, a new proxy problem will be generated to hopefully address the issue. Empirically, as we have demonstrated in the experiments, such variant in the planning behavior results in generally significant improvements in terms of generalization abilities at the cost of extra computation.

### E.1.9 HYPERPARAMETER TUNING

Some hyperparameters introduced by **Skipper** can be located in the pseudocode in Alg. 1.

**Timeout and Pruning Threshold** Intuitively, we tied the timeout to be equal to the distance pruning threshold. The timeout kicks in when the agent thinks a checkpoint can be achieved within *e.g.* $8$ steps, but already spent $8$ steps yet still could not achieve it.

This leads to how we tuned the pruning (distance) threshold: we fully used the advantage of our experiments on DP-solvable tasks: with a snapshot of the agent during its training, we can sample many $\langle$ starting state, target state $\rangle$ pairs and calculate the ground truth distance between the pair, as well as the failure rate of reaching from the starting state to the target state given the current policy $\pi$, then plot them as the $x$ and $y$ values respectively for visualization. We found such curves to evolve from high failure rate at the beginning, to a monotonically increasing curve, where at small true distances, the failure rates are near zero. We picked $8$ because the curve starts to grow explosively when the true distances are more than $9$.

$k$ **for** $k$**-medoids** We tuned this by running a sensitivity analysis on **Skipper** agents with different $k$'s, whose results are presented previously in this Appendix.

Additionally, we prune from 32 checkpoints because 32 checkpoints could achieve (visually) a good coverage of the state space as well as its friendliness to NVIDIA accelerators.

**Size of local Perception Field** We used a local perception field of size 8 because our baseline model-free agent would be able to solve and generalize well within $8 \times 8$ tasks, but not larger. Roughly speaking, our spatial abstraction breaks down the overall tasks into $8 \times 8$ sub-tasks, which the policy could comfortably solve.

**Model-free Baseline Architecture** The baseline architecture (distributional, Double DQN) was heavily influenced by the architecture used in the previous work (Zhao et al., 2021), which demonstrated success on similar but smaller-scale experiments ($8 \times 8$). The difference is that while then we used computationally heavy components such as transformer layers on a set-based representation, we replaced them with a simpler and effective local perception component. We validated our model-free baseline performance on the tasks proposed in Zhao et al. (2021).

## E.2 LEAP

### E.2.1 ADAPTATION FOR DISCRETE ACTION SPACES

The LEAP baseline has been implemented from scratch for our experiments, since the original open-sourced implementation[4] was not compatible with environments with discrete action spaces. LEAP's training involves two pretraining stages, that are, generator pretraining and distance estimator pretraining, which were originally named the VAE and RL pretrainings. Despite our best effort, that is to be covered in detail, we found that LEAP was unable to get a reasonable performance in its original form after rebasing it on a discrete model-free RL baseline.

### E.2.2 REPLACING THE MODEL

We tried to identify the reasons why the generalization performance of the adapted LEAP was unsatisfactory: we found that the original VAE used in LEAP is not capable to handle even few training tasks, let alone generalize well to the evaluation tasks. Even by combining the idea of the context / partial description split (still with continuous latents), during decision time, the planning results given by the evolutionary algorithm (Cross Entropy Method, CEM, Rubinstein (1997a)) almost always produce delusional plans that are catastrophic in terms of performance. This was why we switched into LEAP the same conditional generator we proposed in the paper, and adapted CEM accordingly, due to the change from continuous latents to discrete.

We also did not find that using the pretrained VAE representation as the state representation during the second stage helped the agent's performance, as the paper claimed. In fact, the adapted LEAP variant could only achieve decent performance after learning a state representation from scratch in the RL pretraining phase. Adopting **Skipper**'s splitting generator also disables such choice.

### E.2.3 REPLACING TDM

The original distance estimator based on Temporal Difference Models (TDM) also does not show capable performance in estimating the length of trajectories, even with the help of a ground truth distance function (calculated with DP). Therefore, we switched to learning the distance estimates with our proposed method. Our distance estimator is not sensitive to the sub-goal time budget as TDM and is hence more versatile in environments like that was used in the main paper, where the trajectory length of each checkpoint transition could highly vary. Like for **Skipper**, an additional terminal estimator has been learned to make LEAP planning compatible with the terminal lava states. Note that this LEAP variant was trained on the same sampling scheme with HER as in **Skipper**.

The introduced distance estimator, as well as the accompanying full-state encoder, are of the same architecture, hyperparameters, and training method as those used in **Skipper**. The number of intermediate subgoals for LEAP planning is tuned to be 3, which close to how many intermediate checkpoints **Skipper** typically needs to reach before finishing the tasks. The CEM is called with 5 iterations for each plan construction, with a population size of 128 and an elite population of size 16. We found no significant improvement in enlarging the search budget other than additional wall

---

[4]https://github.com/snasiriany/leap

time. The new initialization of the new population is by sampling a $\epsilon$-mean of the elite population (the binary partial descriptions), where $\epsilon = 0.01$ to prevent the loss of diversity. Because of the very expensive cost of using CEM at decision time and its low return of investment in terms of generalization performance, during the RL pretraining phase, the agent performs random walks over uniformly random initial states to collect experience.

### E.2.4 FAILURE MODE: DELUSIONAL PLANS

Interestingly, we find that a major reason why LEAP does not generalize well is that it often generates delusional plans that lead to catastrophic subgoal transitions. This is likely because of its blind optimization in the latent space towards shorter path plans: any paths with delusional shorter distances would be preferred. We present the results with LEAP combined with our proposed delusion suppression technique in Fig. 8. We find that the adapted LEAP agent, with our generator, our distance estimator, and the delusion suppression technique, is actually able to achieve significantly better generalization performance.

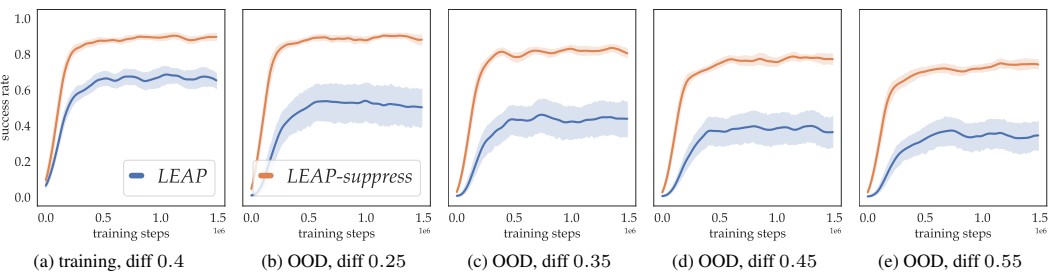

(a) training, diff 0.4  (b) OOD, diff 0.25  (c) OOD, diff 0.35  (d) OOD, diff 0.45  (e) OOD, diff 0.55

Figure 8: **Comparative Results of LEAP with and without the delusion suppression technique**: the results are obtained with 50 training tasks. The results are obtained from 20 independent seed runs.

### E.3 DIRECTOR

### E.3.1 ADAPTATION

We based our experiments of Director (Hafner et al., 2022) on the publicly available code (`https://github.com/danijar/director`) released by the authors. Except for a few changes in the parameters, which are depicted in Tab. 1, we have used the default configuration provided for Atari environments. Note that as the Director version in which the worker receives no task rewards performed worse in our tasks, we have used the version in which the worker receives scaled task rewards (referred to as "Director (worker task reward)" in Hafner et al. (2022)). This agent has also been shown to perform better across various domains in Hafner et al. (2022).

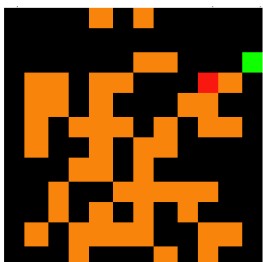

Figure 9: **An example for simplified observations for Director.**

**Encoder.** Unlike **Skipper** and LEAP agents, the Director agent receives as input a simplified RGB image of the current state of the environment (see Fig. 9). This is because we found that Director performed better with its original architecture, which was designed for image-based observations. We removed all textures to simplify the RGB observations.

### E.3.2 FAILURE MODES: BAD GENERALIZATION, SENSITIVE TO SHORT TRAJECTORIES

**Training Performance.** We investigated why Director is unable to achieve good training performance(Fig. 3). As Director was designed to be trained solely on environments where it is able to collect long trajectories to train a good enough recurrent world model (Hafner et al., 2022), we hypothesized that Director may perform better in domains where it is able to interact with the environment through longer trajectories by having better recurrent world models (*i.e.*, the agent does not

Table 1: The changed parameters and their values in the config file of the Director agent.

| Parameter | Value |
|---|---|
| replay_size | 2M |
| replay_chunk | 12 |
| imag_horizon | 8 |
| env_skill_duration | 4 |
| train_skill_duration | 4 |
| worker_rews | {extr: 0.5, expl: 0.0, goal: 1.0} |
| sticky | False |
| gray | False |

immediately die as a result of interacting with specific objects in the environment). To test this, we experimented with variants of the used tasks, where the lava cells are replaced with wall cells, so the agent does not die upon trying to move towards them (we refer to this environment as the "walled" environment). The corresponding results on 50 training tasks are depicted in Fig. 10. As can be seen, the Director agent indeed performs better within the training tasks than in the environments with lava.

**Generalization Performance.** We also investigated why Director is unable to achieve good generalization (Fig. 3). As Director trains its policies solely from the imagined trajectories predicted by its learned world model, we believe that the low generalization performance is due to Director being unable to learn a good enough world model that generalizes to the evaluation tasks. The generalization performances in both the "walled" and regular environments, depicted in Fig. 10, indeed support this argument. Similar to what we did in the main paper, we also present experimental results for how the generalization performance changes with the number of training environments. Results in Fig. 11 show that the number of training environments has little effect on its poor generalization performance.

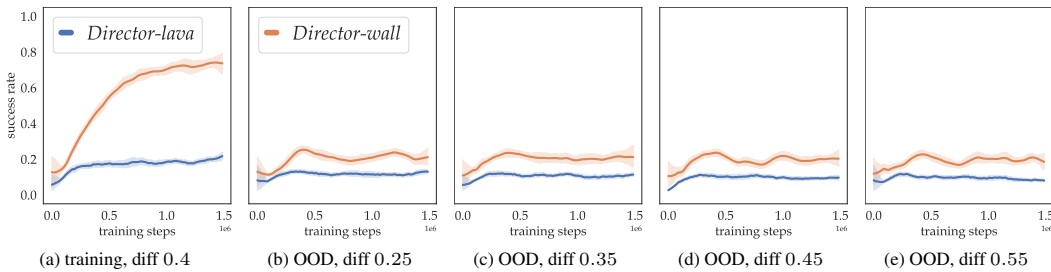

(a) training, diff 0.4   (b) OOD, diff 0.25   (c) OOD, diff 0.35   (d) OOD, diff 0.45   (e) OOD, diff 0.55

Figure 10: **Comparative Results of Director on Environments with Lavas and on those with Walls**: the results are obtained with 50 training tasks. The results for Director-lava (same as in the main paper) are obtained from 20 independent seed runs.

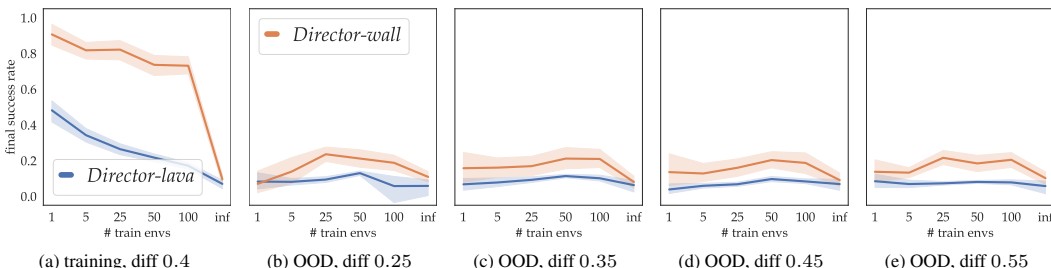

(a) training, diff 0.4   (b) OOD, diff 0.25   (c) OOD, diff 0.35   (d) OOD, diff 0.45   (e) OOD, diff 0.55

Figure 11: **Generalization Performance of Agents on Different Numbers of Training Tasks (while Director runs on the walled environments)**: besides **Director**, each data point and corresponding error bar (95% confidence interval) are processed from the final performance from 20 independent seed runs. **Director**-wall's results are obtained from 20 runs.

## F EXPERIMENTAL RESULTS (CONT.)

We present the experimental results that the main paper could not hold due to the page limit.

### F.1 Skipper-ONCE SCALABILITY

We present the performance of **Skipper-once** on different numbers of training tasks in Fig. 12.

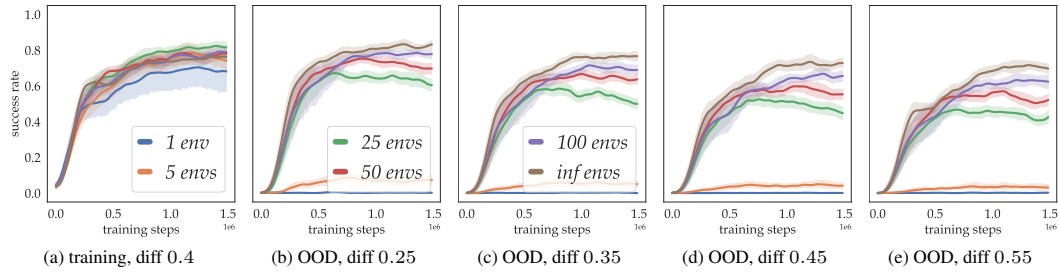

Figure 12: **Generalization Performance of Skipper-once on different numbers of training tasks**: each error bar (95% confidence interval) is obtained from 20 independent seed runs.

### F.2 Skipper-REGEN SCALABILITY

We present the performance of **Skipper-regen** on different numbers of training tasks in Fig. 13.

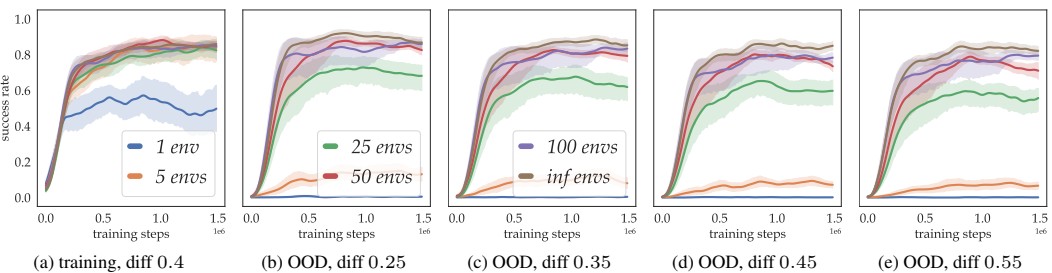

Figure 13: **Performance of Skipper-regen on different numbers of training tasks**: each error bar (95% confidence interval) is obtained from 20 independent seed runs.

### F.3 MODELFREE BASELINE SCALABILITY

We present the performance of the **modelfree** baseline on different numbers of training tasks in Fig. 14.

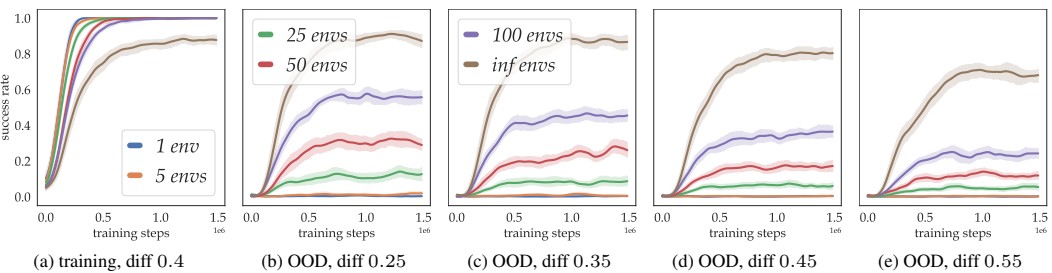

Figure 14: **Generalization Performance of the modelfree baseline on different numbers of training tasks**: each error bar (95% confidence interval) is obtained from 20 independent seed runs.

## F.4 LEAP SCALABILITY

We present the performance of the adapted LEAP baseline on different numbers of training tasks in Fig. 15.

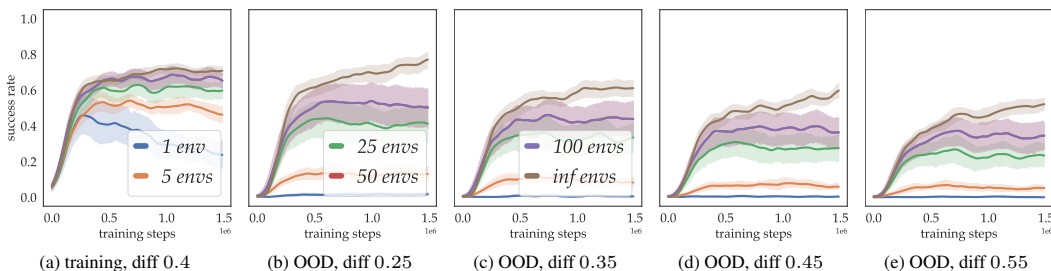

Figure 15: **Generalization Performance of the LEAP baseline on different numbers of training tasks**: each error bar (95% confidence interval) is obtained from 20 independent seed runs.

## F.5 DIRECTOR SCALABILITY

We present the performance of the adapted Director baseline on different numbers of training tasks in Fig. 16.

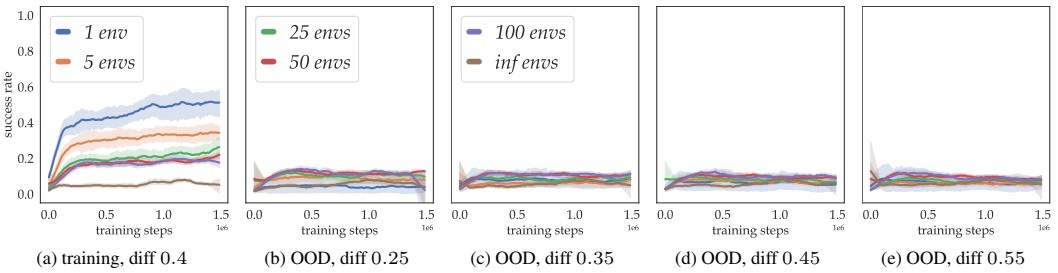

Figure 16: **Generalization Performance of the Director baseline on different numbers of training tasks**: each error bar (95% confidence interval) is obtained from 20 independent seed runs.

## F.6 GENERALIZATION PERFORMANCE ON DIFFERENT NUMBERS OF TRAINING TASKS

The performance of all agents on all training configurations, *i.e.* different numbers of training tasks, are presented in Fig. 17, Fig. 18, Fig. 19, Fig. 20, Fig. 21 and Fig. 22.

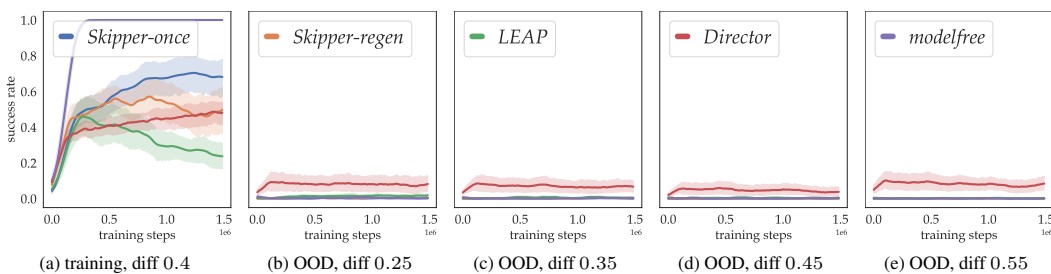

Figure 17: **Generalization Performance of the Agents when trained with 1 training task**: each error bar (95% confidence interval) is obtained from 20 independent seed runs.

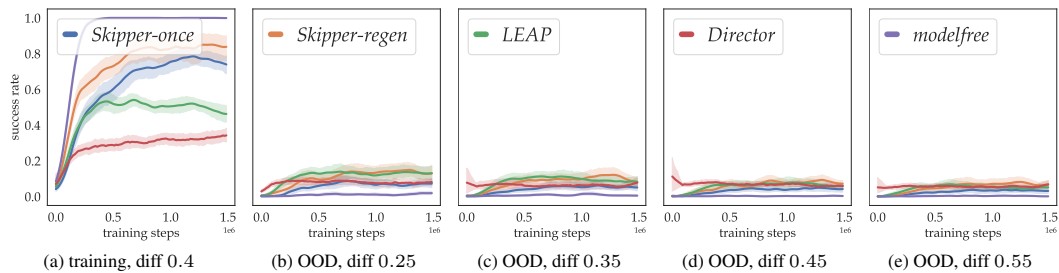

Figure 18: **Generalization Performance of the Agents when trained with** 5 **training tasks**: each error bar (95% confidence interval) is obtained from 20 independent seed runs.

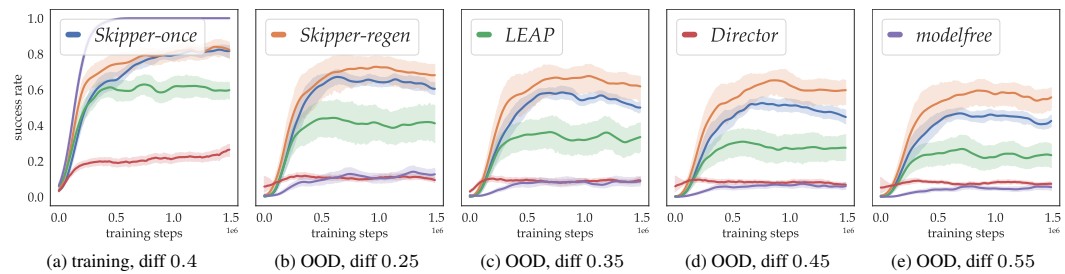

Figure 19: **Generalization Performance of the Agents when trained with** 25 **training tasks**: each error bar (95% confidence interval) is obtained from 20 independent seed runs.

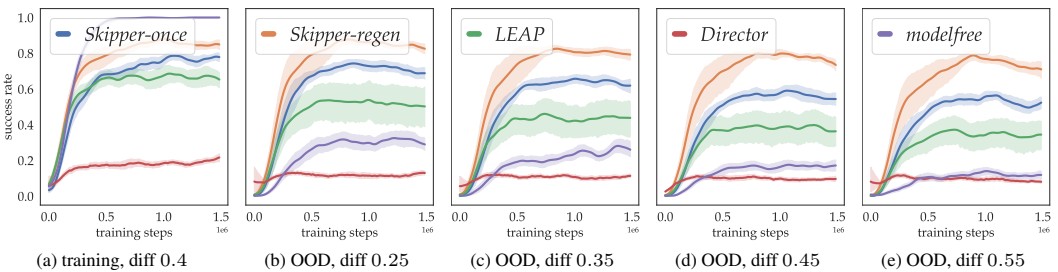

Figure 20: **Generalization Performance of the Agents when trained with** 50 **training tasks (same as in the main paper)**: each error bar (95% confidence interval) is obtained from 20 independent seed runs.

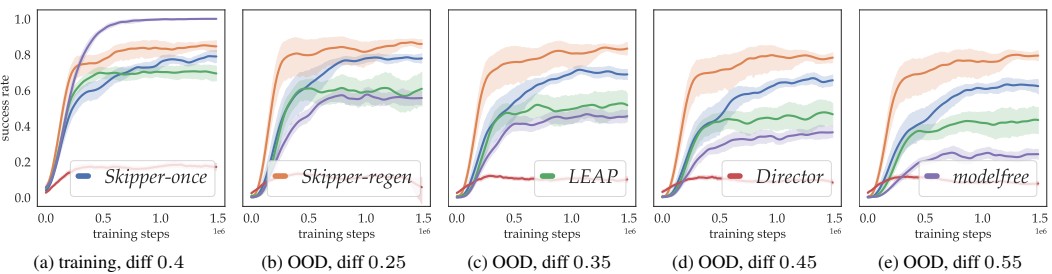

Figure 21: **Generalization Performance of the Agents when trained with** 100 **training tasks**: each error bar (95% confidence interval) is obtained from 20 independent seed runs.

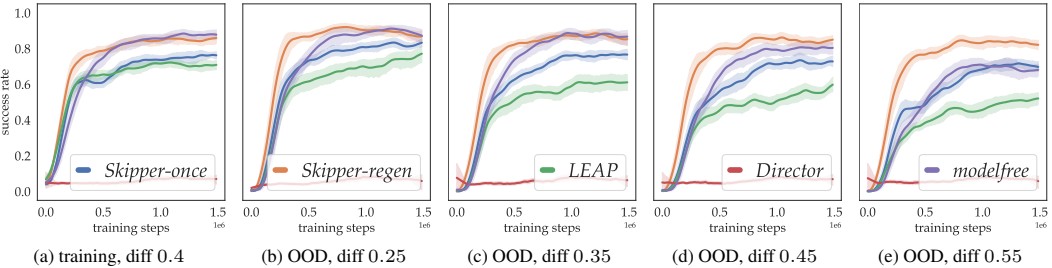

(a) training, diff 0.4     (b) OOD, diff 0.25     (c) OOD, diff 0.35     (d) OOD, diff 0.45     (e) OOD, diff 0.55

Figure 22: **Generalization Performance of the Agents when trained with ∞ training tasks (a new task each training episode)**: each error bar (95% confidence interval) is obtained from 20 independent seed runs.

### F.6.1  STATISTICAL SIGNIFICANCE & POWER ANALYSES

Besides visually observing generally non-overlapping confidence intervals, we present the pairwise $t$-test results of **Skipper-once** and **Skipper-regen** against the compared methods. In addition, if the advantage is significant, we perform power analyses to determine if the number of seed runs (20) was enough to make the significance claim. These results are shown in Tab. 2 and Tab. 3, respectively.

Table 2: **Skipper-once** *v.s.* others: significance & power

|  | method \task difficulty | 0.25 | 0.35 | 0.45 | 0.55 |
|---|---|---|---|---|---|
| 1 train envs | leap | **22** | **NO** | **NO** | **NO** |
|  | director | 15 | 11 | **22** | 11 |
|  | baseline | **NO** | **38** | **36** | **NO** |
| 5 train envs | leap | **28** | **NO** | **NO** | **NO** |
|  | director | **NO** | **NO** | **NO** | **22** |
|  | baseline | 11 | 8 | 10 | 12 |
| 25 train envs | leap | 15 | 13 | 11 | 7 |
|  | director | 2 | 2 | 2 | 2 |
|  | baseline | 2 | 2 | 2 | 2 |
| 50 train envs | leap | 17 | 16 | 11 | 11 |
|  | director | 2 | 2 | 2 | 2 |
|  | baseline | 2 | 2 | 2 | 2 |
| 100 train envs | leap | 15 | 10 | 7 | 9 |
|  | director | 2 | 2 | 2 | 2 |
|  | baseline | 2 | 2 | 2 | 2 |
| inf train envs | leap | **32** | 5 | 7 | 3 |
|  | director | 2 | 2 | 2 | 2 |
|  | baseline | **NO** | **NO** | **NO** | **NO** |

$t$ threshold: 0.05.

Effect size set to be the difference of the means of the compared pairs (Colas et al., 2018).

Cells are **bold** if results **NOT significant** or **insufficient seeds for statistical power**.

For significant cases, the minimum number of seeds for statistical power 0.2 is provided.

As we can observe from the tables, generally there is significant evidence of generalization advantage in **Skipper** variants compared to the other methods, especially when the number of training environments are between 25 to 100. Additionally, as expected, **Skipper-regen** displays more dominating performance compared to that of **Skipper-once**.

Table 3: **Skipper-regen** *v.s.* others: significance & power

| | method \task difficulty | 0.25 | 0.35 | 0.45 | 0.55 |
|---|---|---|---|---|---|
| 1 train envs | leap | **32** | **NO** | **NO** | **NO** |
| | director | 16 | 13 | **23** | 10 |
| | baseline | **NO** | **NO** | **NO** | **NO** |
| 5 train envs | leap | **NO** | **NO** | **NO** | **NO** |
| | director | **33** | **NO** | **NO** | **NO** |
| | baseline | 6 | 8 | 4 | 5 |
| 25 train envs | leap | 10 | 7 | 5 | 4 |
| | director | 2 | 2 | 2 | 2 |
| | baseline | 2 | 2 | 2 | 2 |
| 50 train envs | leap | 6 | 4 | 3 | 3 |
| | director | 2 | 2 | 2 | 2 |
| | baseline | 2 | 2 | 2 | 2 |
| 100 train envs | leap | 7 | 3 | 3 | 2 |
| | director | 2 | 2 | 2 | 2 |
| | baseline | 2 | 2 | 2 | 2 |
| inf train envs | leap | 15 | 3 | 2 | 2 |
| | director | 2 | 2 | 2 | 2 |
| | baseline | **NO** | **NO** | **35** | 5 |

$t$ threshold: $0.05$.
Effect size set to be the difference of the means of the compared pairs (Colas et al., 2018).
Cells are **bold** if results **NOT significant** or **insufficient seeds for statistical power**.
For significant cases, the minimum number of seeds for statistical power $0.2$ is provided.

# G ABLATION & SENSITIVITY

## G.1 VALIDATION OF EFFECTIVENESS ON STOCHASTIC ENVIRONMENTS

We present the performance of the agents in stochastic variants of the used environment. Specifically, with probability $0.1$, each action is changed into a random action. We present the $50$-training tasks performance evolution in Fig. 23. The results validate the compatibility of our agents with stochasticity in environmental dynamics. Notably, the performance of the baseline deteriorated to worse than even Director with the injected stochasticity. The compatibility of Hierarchical RL frameworks to stochasticity has been investigated in Hogg et al. (2009).

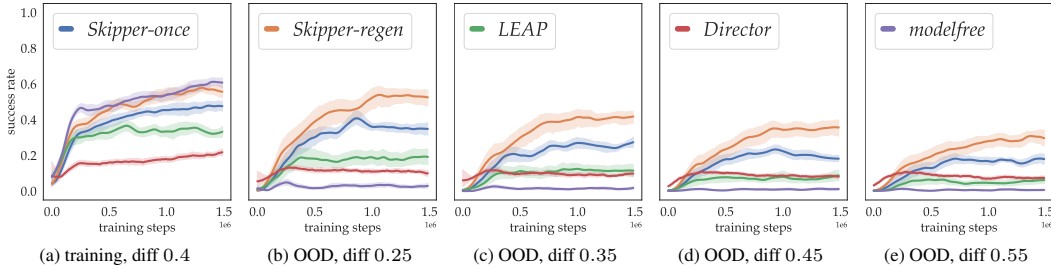

(a) training, diff 0.4    (b) OOD, diff 0.25    (c) OOD, diff 0.35    (d) OOD, diff 0.45    (e) OOD, diff 0.55

Figure 23: **Generalization Performance of agents in stochastic environments**: $\epsilon$-greedy style randomness is added to each primitive action with $\epsilon = 0.1$. Each agent is trained with 50 environments and each curve is processed from 20 independent seed runs.

## G.2 ABLATION FOR SPATIAL ABSTRACTION

We present in Fig. 24 the ablation results on the spatial abstraction component with **Skipper-once** agent, trained with $50$ tasks. The alternative component of the attention-based bottleneck, which is without the spatial abstraction, is an MLP on a flattened full state. The results confirm significant

advantage in terms of generalization performance as well as sample efficiency in training, introduced by spatial abstraction.

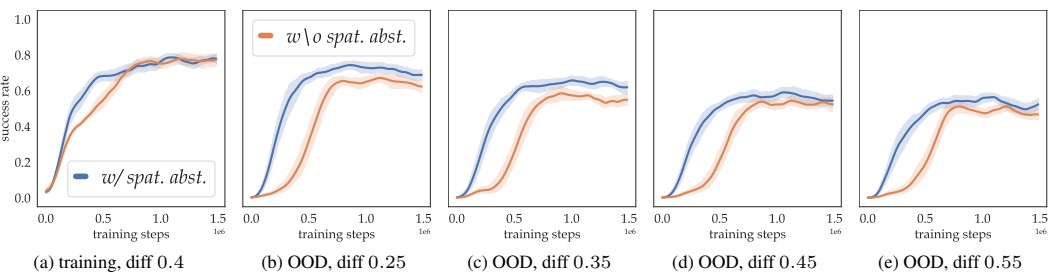

Figure 24: **Ablation for Spatial Abstraction on Skipper-once agent**: each agent is trained with 50 environments and each curve is processed from 20 independent seed runs.

## G.3 ACCURACY OF PROXY PROBLEMS & CHECKPOINT POLICIES

We present in Fig. 25 the ablation results on the accuracy of proxy problems as well as the checkpoint policies of the **Skipper-once** agents, trained with 50 tasks. The ground truths are computed via DP on the optimal policies, which are also suggested by DP. Concurring with our theoretical analyses, the results indicate that the performance of **Skipper** is determined (bottlenecked) by the accuracy of the proxy problem estimation on the high-level and the optimality of the checkpoint policy on the lower level. Specifically, the curves for the generalization performance across training tasks, as in (a) of 25, indicate that the lower than expected performance is a composite outcome of errors in the two levels. In the next part, we address a major misbehavior of inaccurate proxy problem estimation - chasing delusional targets.

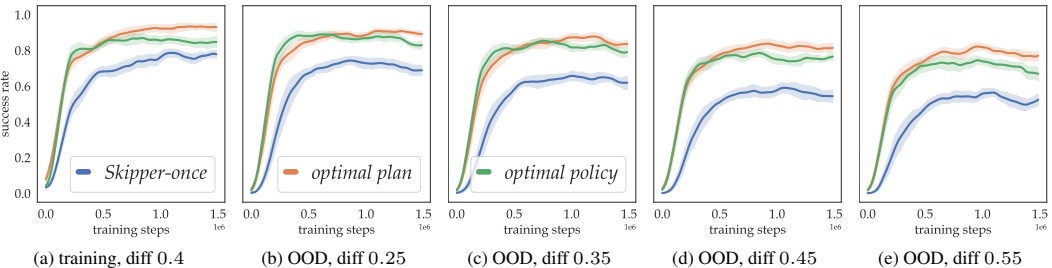

Figure 25: **Skipper-once Empirical Performance *v.s.* ground truths**: both the optimal policy and optimal plan variants are assisted by DP. The default deterministic setting induces the fact that combining optimal policy and optimal plan results in 1.0 success rate. The figures suggest that the learned agent is limited by errors both in the proxy problem estimation and the checkpoint policy $\pi$. Each agent is trained with 50 environments and each curve is processed from 20 independent seed runs.

## G.4 TRAINING INITIALIZATION: UNIFORM V.S. SAME AS EVALUATION

We compare the agents' performance with and without uniform initial state distribution. The non-uniform starting state distributions introduce additional difficulties in terms of exploration. In Presented in Fig. 26, these results are obtained from training on 50 tasks. We conclude that given similar computational budget, using non-uniform initialization only slows down the learning curves without introducing significant changes to our conclusions, and thus we use the ones with uniform initialization for presentation in the main paper.

## G.5 ABLATION: VERTEX PRUNING

As mentioned previously, each proxy problem in the experiments are reduced from 32 vertices to 12 with such techniques. We compare the performance curves of the used configuration against

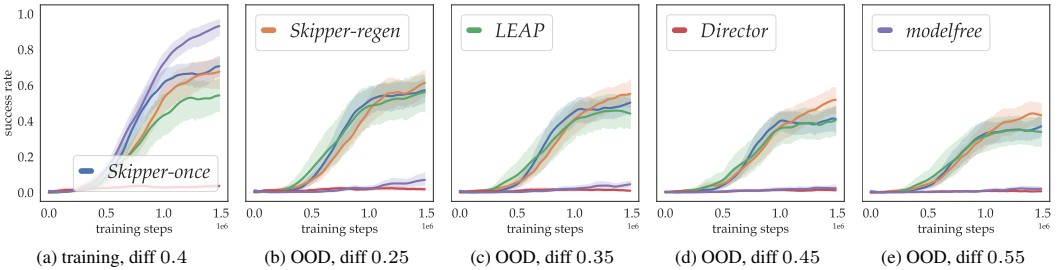

Figure 26: **Comparative Results on** 50 **training tasks without uniform initial state distribution**: each curve is processed from 20 independent seed runs.

a baseline that generates 12-vertex proxy problems without pruning. We present in Fig. 27 these ablation results on the component of $k$-medoids checkpoint pruning. We observe that the pruning not only increases the generalization but also the stability of performance.

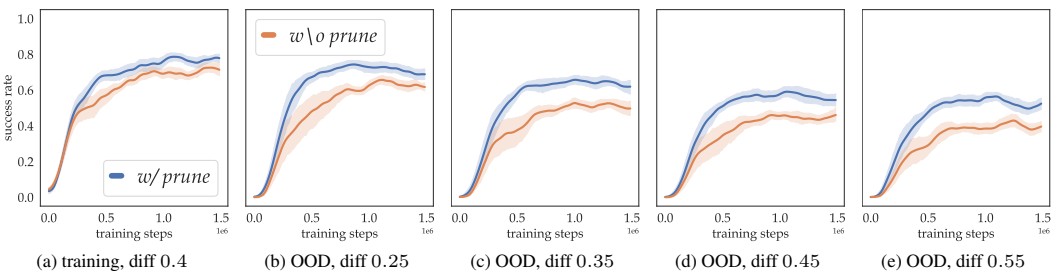

Figure 27: **Ablation Results on** 50 **training tasks for** $k$-**medoids pruning**: each curve is processed from 20 independent seed runs.

### G.6 SENSITIVITY: NUMBER OF VERTICES

We provide a sensitivity analysis to the number of checkpoints (number of vertices) in each proxy problem. We present the results of **Skipper-once** on 50 training tasks with different numbers of post-pruning checkpoints (all reduced from 32 by pruning), in Fig. 28. From the results, we can see that as long as the number of checkpoints is above 6, **Skipper** exhibits good performance. We therefore chose 12, the one with a rather small computation cost, as the default hyperparameter.

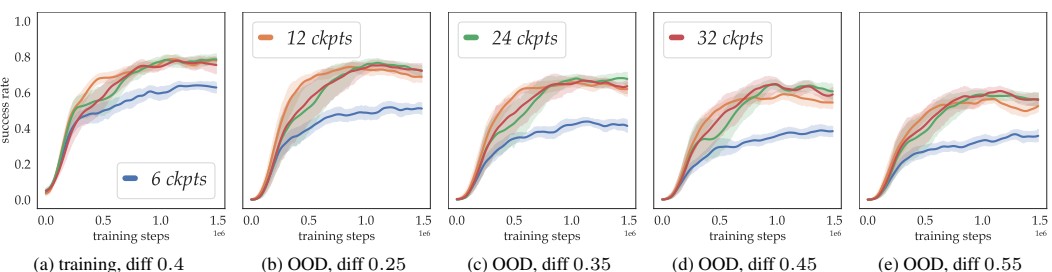

Figure 28: **Sensitivity of Skipper-once on the number of checkpoints in each proxy problem**: each agent is trained with 50 environments. All curves are processed from 20 independent seed runs.

### G.7 ABLATION: PLANNING OVER PROXY PROBLEMS

We provide additional results for the readers to intuitively understand the effectiveness of planning over proxy problems. This is done by comparing the results of **Skipper-once** with a baseline

**Skipper**-goal that blindly selects the task goal as its target all the time. We present the results based on 50 training tasks in Fig. 29. Concurring with our vision on temporal abstraction, we can see that solving more manageable sub-problems leads to faster convergence. The **Skipper**-goal variant catches up later when the policy slowly improves to be capable of solving longer distance navigation.

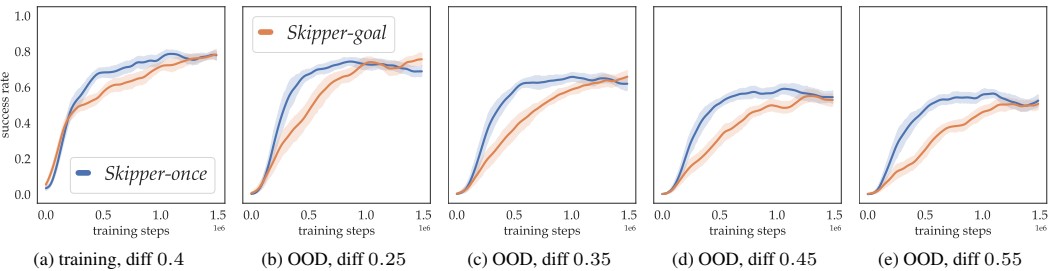

(a) training, diff 0.4   (b) OOD, diff 0.25   (c) OOD, diff 0.35   (d) OOD, diff 0.45   (e) OOD, diff 0.55

Figure 29: **Effectiveness of Proxy Problem based Planning**: each agent is trained with 50 environments and each curve is processed from 20 independent seed runs.

