# OpenReview forum: "Consciousness-Inspired Spatio-Temporal Abstractions for Better Generalization in Reinforcement Learning"
_ICLR.cc/2024/Conference — ICLR 2024 poster_

### Official Review · Reviewer_qHoq · 2023-10-29

**Soundness:** 4 excellent
**Presentation:** 3 good
**Contribution:** 3 good
**Rating:** 6
**Confidence:** 2

**Summary:**

The paper introduces a planning agent called Skipper that overcomes the limitations of existing reinforcement learning (RL) agents.

RL agents can either rely on intuition or only plan for short-term tasks. Skipper, on the other hand, uses human-like thinking to break down complex tasks into smaller parts called "proxy" problems.

These problems are represented as graphs, with decision points as dots and connections between them as lines. Skipper solves each problem and then focuses on the connections to solve the next part.

This approach helps Skipper handle new situations better and create useful shortcuts. The paper also shows that Skipper performs better than other methods in terms of adapting to new situations, even with limited training.

**Strengths:**

Skipper employs both temporal and spatial abstraction. Temporal abstraction breaks down complex tasks into smaller sub-tasks, while spatial abstraction allows the agent to focus on relevant aspects of the environment, improving learning efficiency and generalization.

Decision-time planning enhances the agent's ability to handle novel situations. By formulating a simplified proxy problem and planning within it, the agent can generate plans in real time, adapting to changing environments or goals.

Skipper uses proxy problems, represented as finite graphs, to create a simplified version of the task. The use of checkpoints as vertices in the proxy problems allows for a sparse representation, reducing computational complexity.

The experiments are conducted in gridworld environments with varying levels of difficulty, providing a realistic and challenging testbed for the agents. The use of MiniGrid-BabyAI framework ensures a well-defined and computationally feasible evaluation environment.

**Weaknesses:**

The experiments are conducted in simulated grid world environments, which might not fully represent the complexities of real-world scenarios. The findings might not directly translate to applications in more intricate, dynamic, and unstructured environments.

In Section 4: the paper assumes fully observable environments, which might not hold in many real-world applications where agents have limited or partial observability. The performance of Skipper in partially observable environments is not explored, limiting its practical applicability.

The paper introduces different variants of Skipper. While this provides a comprehensive analysis, it also complicates the understanding of the core method. A clearer presentation of the differences and justifications for each variant would enhance the paper's clarity.

**Questions:**

What are the potential future directions for research and development in the field of planning agents like Skipper? Are there any specific improvements or enhancements that you plan to explore in your future work? For example, how does this method influence the research of more complex environments?

Can you provide more information about the theoretical guarantees of the quality of the solution to the overall problem? What are the key insights or findings from the theoretical analysis? For example, what are  the limitations when applying Skipper in the continuous control environments.

The authors mentioned that non-uniform starting state distributions introduce additional difficulties in terms of exploration and therefore globally slow down the learning process. Is this caused by the intrinsic limitations of the Skipper method?

**Details Of Ethics Concerns:**

No ethics concerns since it focuses on Grid-world studies and RL algorithms only.

---

> ### Author Response · Authors · 2023-11-19
> **(1/3) Point-by-point Response to Reviewer qHoq's comments**
>
> (1/3)
>
>
> **Thank you very much for your comments! Here are our point-by-point replies to your concerns.**
>
> * The experiments are conducted in simulated grid world environments, which might not fully represent the complexities of real-world scenarios. The findings might not directly translate to applications in more intricate, dynamic, and unstructured environments.
>
> *We acknowledge the limitations of the scope of our experiments and would like to continue to investigate in richer more general environmental settings that could translate directly to real-world applications. That said, we would like to emphasize that the key motivation of our work, and what this work seeks to achieve, that is - a sound foundational model-based framework that can lead to better generalization in longer-term planning, future works that may be directly translatable for application. Its insights serve as the first step towards a truly generalizable, real-world applicable approach.*
>
> * In Section 4: the paper assumes fully observable environments, which might not hold in many real-world applications where agents have limited or partial observability. The performance of Skipper in partially observable environments is not explored, limiting its practical applicability.
>
> *We acknowledge that we limited our experiments to only fully observable cases for simplicity and our limited academic computational resources. However, we would point out that Skipper framework is not incompatible with any recurrent / memory-based components that can be embedded to handle partial observability. We have high confidence that in our continued research (on trying to extend to procgen), partial observability would be handled appropriately by us with more computational resources.*
>
> * The paper introduces different variants of Skipper. While this provides a comprehensive analysis, it also complicates the understanding of the core method. A clearer presentation of the differences and justifications for each variant would enhance the paper's clarity.
>
> *Thank you very much for your suggestion. We originally put a concise version of the differences between the two variants in the beginning of section 4 due to page limits and overwhelming amount of content. Following your suggestions, we have now added an extended section to discuss and clarify the two variants in a separate section in the Appendix.*
>
> > The only difference between the two variants, i.e. Skipper-once and Skipper-regen is that the latter variant would discard the previously constructed proxy problem and construct a new one every time the planning is triggered. This introduces more computational effort while lowering the chance that the agent gets ``trapped in a bad proxy problem that cannot form effective plans to achieve the goal. If such a situation occurs with Skipper-regen, as long as the agent does not terminate the episode prematurely, a new proxy problem will be generated to hopefully address the issue. Empirically, as we have demonstrated in the experiments, such variant in the planning behavior results in generally significant improvements in terms of generalization abilities at the cost of extra computation.
>
> * What are the potential future directions for research and development in the field of planning agents like Skipper? Are there any specific improvements or enhancements that you plan to explore in your future work? For example, how does this method influence the research of more complex environments?
>
> ***Future directions for the field:***
>
> *Skipper is motivated to investigate insights into how RL agents can generalize better out-of-distribution such that RL could be used in real-world applications to help advance humanity. Planning agents, in our opinion, should reflect on their incapability to plan meaningfully, where the most difficulty lies in inappropriate abstractions of decision-making, that Skipper seeks to address.*
>
> ***Future directions for Skipper:***
>
>  - *Improve the generation of random checkpoints to meaningful checkpoints: as discussed in Section 6. This would be an advancement on the foundational front to understand the limitations and potentials of this framework;*
>
>  - *Combine Skipper with a traditional model-free agent to assist its long-term reasoning capabilities: because of the off-policy compatibility, the whole Skipper framework can be used as an auxiliary learner alongside an RL agent. The auxiliary Skipper can be called during inference for better generalization.*
>
>  - *Validate and scale up the experiments on environments with more complicated features: this would be to validate how Skipper, currently in a form of simplistic prototype, can combine with the existing techniques to handle richer and more complicated observations in potentially partially observable settings as well as continuous action/state spaces.*
>
> ***(this reply is to be continued in the next part of the response)***

---

> ### Author Response · Authors · 2023-11-19
> **(2/3) Point-by-point Response to Reviewer qHoq's comments**
>
> (2/3)
>
> ***(continuing the last reply)***
>
> *Specifically, for your question regarding the influence of this work for more complex environments: this paper acts as a reminder that the complexity of environments does not lie only in the richness of the observation space but also the intrinsic reasoning difficulties of tasks (e.g. planning for long term). This generalization-focused work should also bring healthier focus on the generalization abilities, i.e. the applicability of RL, instead of overfitting on boosting highest scores on popular benchmarks. Based on the ground truths acquired by solving the tasks with dynamic programming, this paper uses detailed and accurate measures to validate the effectiveness of its components. This is an irreplaceable advantage compared to the popular benchmarks that focus primarily on acquiring scores in environments with complex observations, instead of meaningful aspects such as generalization.*
>
> * Can you provide more information about the theoretical guarantees of the quality of the solution to the overall problem? What are the key insights or findings from the theoretical analysis? For example, what are the limitations when applying Skipper in the continuous control environments.
>
> *In the submission, we already provided theoretical insights of overall performance, which is later validated by our experiments. In addition to those, despite trying hard during the rebuttal stage, we could not offer a responsible claim to new discoveries. Nevertheless, we would like to expand the implications of the theoretical results here as well (also added to the paper Appendix):*
>
> > We do not know if our current theory truly captures the bounds of the full applicability of our framework, we also couldn’t discover and investigate Skipper’s empirical performance in all the theoretically sound cases. We would gladly resume our journey for deeper understanding.
>
> > Theorem 1 provides guarantees on the solution to the overall problem. The quality of the solution depends on both the quality of the estimates (distances/discounts, rewards) as well as the quality of the policy, as the theorem guarantees accuracy to the solution of the overall problem given a current policy, which should evolve towards optimal during training. This means bad policy with good estimation will lead to an accurate yet bad overall solution. No matter the quality of the policy, with a bad estimation, it will result in a poor estimate of solutions. Only a near-optimal policy and good estimation will lead to a near-optimal solution.
>
> *Specifically for your question, a continuous action space would not introduce a major change to Skipper’s framework, expect that we need to replace the baseline agent with a compatible one such as TD3. More specifically, the estimations will change from a discrete DQN style S -> R^{|A|} to DDPM style S x A -> R.*
>
> *While if the state space becomes continuous, we may need to (but not necessarily) introduce a certain distance measure to mark the agents’ achievement of a checkpoint. In this case, a proposed checkpoint may never be perfectly fulfilled but a rough equivalence would still be likely to work.*
>
> *For example, in our discrete state space experiments, the achievement of checkpoints can be either identified by comparing the identity between the target checkpoint and the current state, or their partial descriptions (currently used). For continuous state spaces, where a strict identity cannot be established, we either must adopt a distance measure for approximating the equivalence between the target checkpoint and the current state, or we continue to use the equivalence of the partial descriptions. We intentionally designed the partial descriptions to be in the form of bundles of binary variables, so that this comparison could be done fast and trivially for any forms of the state space.*
>
> * The authors mentioned that non-uniform starting state distributions introduce additional difficulties in terms of exploration and therefore globally slow down the learning process. Is this caused by the intrinsic limitations of the Skipper method?
>
> *We are very excited to answer this insightful question!*
>
> *When the non-uniform starting state distribution is adopted, we observe that the Skipper agents maintain a relatively low performance for a long time before its success rate skyrockets.*
>
> *Because we were able to solve the environmental truth with dynamic programming, which is an advantage few other environments could bring, we were able to identify that the limiting factor for its slowdown is indeed the extra exploration difficulty. More specifically, the negligible density of non-zero reward at the early stages of the training. Here is a detailed explanation of why that is the case:*
>
> ***(this reply is to be continued in the next part of the response)***

---

> > ### Author Response · Authors · 2023-11-19
> > **(3/3) Point-by-point Response to Reviewer qHoq's comments**
> >
> > (3/3)
> >
> > ***(continuing the last reply)***
> >
> > *With our non-uniform starting state distribution, where we put the agents’ start location as far as possible away from the goal, the initial exploration stage will save into the experience replay almost no transitions with meaningful reward. This means, most of the transitions are just from state to state without rewards. The reward estimator, which learns from the experience replay, in turn learned to trivially predict that all checkpoint transitions yield 0 cumulative reward (therefore the agent is not really incentivised to go anywhere). The code we submitted contains a periodic calculation of how much the overall estimated rewards deviate from the ground truth, as well as a breakdown of that deviation into the checkpoint transitions that are trivial to predict (0 reward) and non-trivial ones (with reward). We observe every time in tensorboard, the non-trivial reward error (for predicting the rewards for rewarding checkpoints), compared to the ground truth, remains high during the training process until a moment that it drops down dramatically to almost zero, where at the same time the agent’s success rate improve dramatically to converge to its final performance (we invite you to try seeing it yourself). When there are rewarding transitions in the experience replay, since they yield high training error, they will be promoted by the prioritized sampling to rapidly change the predictions of the reward estimator such that meaningful plan would be proposed at decision-time, which in turn results in more rewarding transitions being stored into experience replay, forming a positive loop that accelerates learning greatly.*
> >
> > *Please check the following figures corresponding to 20 seed runs on the non-trivial reward error as well as the training performance here (against the ground truth, for uniform initialization v.s. non-uniform initialization):*
> >
> > *https://github.com/AnonymousAuthors21/ICLR2024-Rebuttal/blob/main/initialization.png*
> >
> > *With the evidence above, we do not consider this to be a significant intrinsic limitation rather a challenge posed by the nature of these difficult tasks.*
> >
> > *We added these insights into the revised manuscript. We hope that our answers addressed your concerns well and we hope that we demonstrated, through our replies our hopefully deep enough understanding of the topic. We would greatly appreciate if you could consider increasing your rating of this paper, as it matters a lot to us!*
> >
> > ===
> >
> > ***Finally, we hope that our answers addressed your concerns well and have improved the manuscript considerably. We would greatly appreciate it if you could consider increasing your rating of this paper, as it matters a lot to us! Thank you very much for your time!***

---

> > > ### Author Response · Authors · 2023-11-22
> > > **Reminder for Discussion**
> > >
> > > As the end of the author-reviewer discussion is approaching, we kindly ask for your review of our revised paper and response. If our revisions have adequately addressed your concerns, we'd greatly appreciate your consideration in adjusting the scores. Should you have any remaining questions or require further clarification, please don't hesitate to let us know. We would be more than happy to answer any further questions.

---

### Official Review · Reviewer_3j7M · 2023-10-31

**Soundness:** 3 good
**Presentation:** 4 excellent
**Contribution:** 2 fair
**Rating:** 6
**Confidence:** 4

**Summary:**

This paper presents Skipper, a model-based reinforcement learning approach that leverages both spatial and temporal abstraction to perform well on problems with sparse rewards. The abstraction takes the form of what the authors refer to as a "proxy problem": a directed graph in which vertices and edges correspond to states and the ability to reliability traverse between them (respectively). These proxy problems are estimated at decision time from learned models, trained end-to-end via hindsight. The authors show the ability of Skipper to perform well on a navigation-oriented gridworld problem [via the MiniGrid-BabyAI framework] and in particular show how Skipper generalizes quickly given only a handful of training tasks, outperforming both model-based and model-free baselines in its ability to generalize.

**Strengths:**

This is a well written paper. It is easy to follow despite the challenging subject material that presents a well-scoped approach to tackling model-based RL for gridworld-like problems. The paper is generally good and, between the main body of the paper and the expansive additional experiments and details in the Appendix, the work shows the effectiveness of Skipper. Using hindsight to train Skipper is well-motivated. The baseline algorithms against which Skipper is compared are appropriate, and it is clear that Skipper outperforms them under multiple different circumstances. In particular, the generalization-focused experiments are interesting and convincing, and show the benefits of the decision-time planning approach that Skipper takes to planning.

**Weaknesses:**

There are two (related) core weakness of the paper: (1) it seems that Skipper is designed primarily with gridworld environments in mind and seems to be relatively limited in what problems it is suited to solve and (2) the paper is missing a clear "Weaknesses" or "Limitations" section, which would help the reader understand when Skipper will be appropriate.

The experiments in the paper focus somewhat narrowly on a single family of gridworld environments. While this is not on its own a problem, there are questions regarding how applicable Skipper is to tasks beyond those shown in the paper. Specifically, Skipper's vertex generation module consists of an attention mechanism over the observation. For the experiments presented in the manuscript, the observations all take the form of overhead views of a map of the environment and so directly correspond to the state space. This means that Skipper is particularly well-suited to work on environments such as these, since vertex proposal works directly on the observation, and (unless I am mistaken in my understanding) arguably Skipper is over-engineered to work on environments such as these.

The appeal of the other model-based reinforcement learning approaches mentioned in the paper, including MuZero and Dreamer (against which Skipper is compared), is that they work in fairly general observation spaces, both capable of taking as input RGB images and learning a model from those. While the paper on the whole generally does an effective job of communicating what assumptions the system makes and how it works, the construction of Skipper seems to implicitly assume a certain type of state or observation representation that limits the applicability of the work in general. If the authors could clarify this point and make explicit these so-far implicit conditions, it would greatly strengthen the paper.

Relatedly, while the paper spends considerable time discussing the ways in which the proposed Skipper approach is effective, it lacks a clear Weaknesses or Limitations section. Such a section, and comments regarding which problems or domains Skipper is best suited, would greatly aid in understanding when Skipper is appropriate. If possible, I would appreciate hearing the authors' thoughts on the size of map or problem that Skipper would be expected to work; I do not think additional experiments are necessary or appropriate, though I am interested to understand if Skipper would succeed on larger occupancy grids and navigation tasks.

**Questions:**

I would like the authors to address what I identify as the two core weaknesses in my comments above. In summary:

1. Could the authors comment on the (implicit) assumptions Skipper makes about the task/environment? What decisions about Skipper's design (e.g., using an attention mechanism over the observations to propose vertices for proxy problem estimation) are core to Skipper's functionality or can be changed without changing the overall approach? On what types of problems is Skipper useful (or not)?
2. Relatedly, a small summary of Limitations or Weaknesses of the proposed approach would help clarify when Skipper should be used, and thus strengthen the paper. If the answers to my question 1 were in the form of such a section, I think it could be effective and helpful for improving clarity.

---

> ### Author Response · Authors · 2023-11-19
> **(1/3) Point-by-point Response to Reviewer 3j7M's comments**
>
> (1/3)
>
> **Thank you very much for your comments! Here are our point-by-point replies to your concerns.**
>
> * The experiments focus somewhat narrowly on a single family of gridworld environments. While this is not on its own a problem, there are questions regarding how applicable Skipper is to tasks beyond those shown. Specifically, Skipper's vertex generation module consists of an attention mechanism over the observation. For the experiments presented, the observations all take the form of overhead views of a map and so directly correspond to the state space. This means that Skipper is particularly well-suited to work on environments such as these, since vertex proposal works directly on the observation, and arguably Skipper is over-engineered to work on environments such as these.
>
> *Thank you for your comments. We would like to acknowledge that we indeed have only a single family of gridworld environments. This is not only due to their appropriate nature to demonstrate the coherence of the empirical behaviors of Skipper with the motivating theory, but also a product of limited page contents as well as computational resources. Despite that we authors could not disclose our identities, we would like to disclose the fact that **this work is purely done in an academic setting** without many computational resources. We genuinely suggest that you could take this as a first step towards a more generally applicable algorithm that can be empirically validated to work well in other environments.*
>
> *Despite that Skipper isn't tested on other environments, there is no evidence that Skipper would have no applicability elsewhere, as it makes little assumptions on the kind of tasks it is meant to solve, and with absolutely no assumption for gridworlds or incompatibility to extend to partially observable scenarios. The checkpoint generation (vertex proposal in your language) was, with consideration, implemented to work on visual observations to take full advantage of the recent progress in conditional (image) generation, guaranteeing its wider applicability. For non-visual environments, the checkpoint generator should be implemented with appropriate architectures that could handle the corresponding representations of observations. We showcased a Skipper implementation for visual inputs due to their dominant popularity in the current RL community.*
>
> ***We would also like to present some evidence for you to kindly reconsider whether Skipper is over-engineered to work on these experiments. In fact, we were aware of the potentials of over-engineering and hence were adopting consciously the design choices to avoid such issue***:
>
> 1. *As you pointed out, we acknowledge that the experiments are done in a fully-observable setting, where no recurrent / memory mechanisms need to be incorporated. We preferred not to have a partially-observable setting since the contributions of this work will not be further validated, nor would more useful insights be excavated; On the other hand, incorporating recurrent / memory mechanisms into the implementations will not only significantly slow down the experiments (excuse us again for our academic setting) but also truly make Skipper more sensitive to hyperparameters and potentially more overfit on these experiments. **Fundamentally, Skipper’s theory and framework is not limited to fully-observable settings as one could augment the encoder with recurrent / memory mechanisms and let the checkpoint generator work on the level of state representations (in our current experiments, working on the observations and the state representations are equivalent, due to full observability)***
>
> 2. *Contrary to what you said, **the vertex generation model doesn’t consist of an attention mechanism**. The attention (spatial abstraction) is only used for the estimators (cumulative value, distance, checkpoint achieving policy). Our vertex generation model is implemented as a simple conditional CVAE with the default hyperparameters that could work well for generic visual inputs (as well as convoluted feature maps, confirmed by recent advances on latent diffusion). In this paper, the observations are compact low-res images. We believe that, besides dealing with partial-observability, adapting Skipper to other visual domains only requires scaling up the architectures, which we had instead intentionally tuned down for acceleration.*
>
> 3. *Skipper’s architecture is simplistic and bear few changes to the pervasively used distributional DDQN. The major changes include the incorporation of the local perception field and a standard CVAE for checkpoint generation. While minor changes include shrinking down the architectures. All other hyperparameters and the training procedures are not changed. The local perception field was implemented minimalistically: a conv layer (no padding) followed by a depth-1 attention. These were intended to maximize its possible applicable scenarios and to avoid overfitting to the showcased tasks.*

---

> ### Author Response · Authors · 2023-11-19
> **(2/3) Point-by-point Response to Reviewer 3j7M's comments**
>
> (2/3)
>
> * "The appeal of the other model-based reinforcement learning approaches mentioned in the paper, including MuZero and Dreamer (against which Skipper is compared), is that they work in fairly general observation spaces, both capable of taking as input RGB images and learning a model from those. While the paper on the whole generally does an effective job of communicating what assumptions the system makes and how it works, the construction of Skipper seems to implicitly assume a certain type of state or observation representation that limits the applicability of the work in general. If the authors could clarify this point and make explicit these so-far implicit conditions, it would strengthen the paper."
>
> *Thank you very much for your suggestions. We have added a whole section on weaknesses and limitations of this work.*
>
> *Contrary to your comments, Skipper indeed operates on image-like spaces. The difference being that in our experiments, we respected the original BabyAI literature to use a low-resolution image as the observation (12x12). Again, we don’t think that we made additional implicit assumptions of a certain type of representation. However, we have added more clarifications regarding this point in the experiment section:*
>
> > Skipper is implemented to be compatible with image inputs. In our experiments, the observations are low-res two-channel images (12x12x2). More details are presented in the Appendix.
>
>
> * "Relatedly, while the paper spends considerable time discussing the ways in which the proposed Skipper approach is effective, it lacks a clear Weaknesses or Limitations section. Such a section, and comments regarding which problems or domains Skipper is best suited, would greatly aid in understanding when Skipper is appropriate. If possible, I would appreciate hearing the authors' thoughts on the size of map or problem that Skipper would be expected to work; I do not think additional experiments are necessary or appropriate, though I am interested to understand if Skipper would succeed on larger occupancy grids and navigation tasks."
>
>
>
> *Thank you very much for your suggestions. We have added a section on weaknesses and limitations accordingly (due to the page limit, it is placed as of now in the Appendix of the revised manuscript):*
>
> > We generate future checkpoints at random by sampling the partial description space. Despite the post-processing such as pruning, the generated checkpoints do not prioritize on the predictable, important states that matter the most to form a meaningful long-term plan. We would like to continue investigating the possibilities along this line;
>
> > The current implementation is intended for pixel input fully-observable tasks with discrete state and action spaces. Such a minimalistic form is because we wish to isolate the unwanted challenges from other factors that are not closely related to the idea of this work, as well as to make the agent as generalist as possible. On the matter of the compatibility with continuous spaces, Skipper is naturally compatible with continuous actions spaces and the only thing we will need to do is to replace the baseline agent with a compatible one such as TD3 (Fujimoto et al., 2018); on the other hand, for continuous state spaces, the identification of the achievement of a checkpoint becomes tricky. This is due to the fact that a strict identity between the current state and the target checkpoint may be ever established, we either must adopt a distance measure for approximate state equivalence, or rely on the equivalence of the partial descriptions (which is adopted in the current implementation). We intentionally designed the partial descriptions to be in the form of bundles of binary variables, so that this comparison could be done fast and trivially for any forms of the state space; for partial observability, despite that no recurrent mechanism has been incorporated in the current implementation, the framework is not incompatible. To implement that, we will need to augment the state encoder with recurrent or memory mechanisms and we need to make  the checkpoint generator directly work over the learned state representations. We acknowledge that future work is needed to verify Skipper’s performance on the popular partially-observable benchmark suites, which requires the incorporation of components to handle partial observability as well as scaling up the architectures for more expressive power;
>
> > We do not yet know the precise boundaries of the motivating theory on proxy problems, since it only indicates performance guarantees on the condition of estimation accuracy, which in turn does not correspond trivially to a set of well-defined problems. We are eager to explore, outside the scope of sparse-reward navigation, how this approach can be used to facilitate better generalization, and at the same time, try to find more powerful theories that guide us better.”
>
> ***(to be continued in the next part of the response)***

---

> ### Author Response · Authors · 2023-11-19
> **(3/3) Point-by-point Response to Reviewer 3j7M's comments**
>
> (3/3)
>
> ***(continuing the last reply)***
>
> *For your additional questions: with the same architecture and hyperparameters, we have tested Skipper on sizes ranging from 8x8 up to 16x16 (with number of (post-pruning) checkpoints scaling linearly from 8 to 16). Among all these experiments, there are always clear advantages in terms of generalization abilities compared to the other agents. We particularly chose 12x12 because of our limited computational budget, as well as a clear presentation of our results: when the sizes of the problems are small, all methods perform relatively well, the advantages of Skipper cannot be convincingly demonstrated (overlapping confidence intervals); this is also the case in which all methods behave poorly when the size is too large. Additionally, the prohibitive computational demand (due to scaling up the architectures as well as the complexity of the planning process, i.e. quadratically with the number of checkpoints) would cost us several months more to finish the experiments.*
>
>
> * "I would like the authors to address what I identify as the two core weaknesses in my comments above. In summary: Could the authors comment on the (implicit) assumptions Skipper makes about the task/environment? What decisions about Skipper's design (e.g., using an attention mechanism over the observations to propose vertices for proxy problem estimation) are core to Skipper's functionality or can be changed without changing the overall approach? On what types of problems is Skipper useful (or not)?"
>
> *Thank you! To expand beyond the newly incorporated assumptions and limitations section:*
>
> *Technical: optionally a well-defined distance measure for identifying the achievement of checkpoints. For example, in our discrete state space experiments, the achievement of checkpoints can be either identified by comparing the identity between the target checkpoint and the current state, or their partial descriptions (currently used). For continuous state spaces, where a strict identity cannot be established, we either must adopt a distance measure for approximating the equivalence between the target checkpoint and the current state, or we continue to use the equivalence of the partial descriptions. We intentionally designed the partial descriptions to be in the form of bundles of binary variables, so that this comparison could be done trivially for any forms of the state space, making the requirement for the state distance measure optional.*
>
> *Theoretical: while Skipper does not explicitly pose assumptions about environments, our motivating theory only guarantees good performance in the cases where the conditions of Theorem 1 could hold. These conditions however do not simply correspond to a simple family of tasks, e.g. sparse-reward navigation, etc. We validated Skipper’s coherence with the theory on sparse-reward navigation, which is a case where the just-mentioned conditions can be met via learning. We do not know if our current theory truly captures the bounds of the applicability of our framework. Potentially, we could find better theory in the future that could lead us comfortably out of these condition requirements. We also couldn’t discover and investigate Skipper’s performance in all the cases that correspond to the existing theory. We would gladly resume our journey for deeper understanding.*
>
> *We have explained in the previous part of the response that the attention mechanism is not used to propose vertices but to edge estimation. In the Appendix, we provided detailed results of the ablation on the key components, including the local perception field supporting their importance. To the best of our knowledge, Skipper is a simplistic agent with no redundancy in its components. In our opinion, the only substantial change there could be for Skipper is that we could potentially replace the current planning scheme, i.e., value iteration, by fancier methods such as an evolutionary algorithm used in LEAP, with the accompanying downsides and risks (explained in the paper).*
>
> * "Relatedly, a small summary of Limitations or Weaknesses of the proposed approach would help clarify when Skipper should be used, and thus strengthen the paper. If the answers to my question 1 were in the form of such a section, I think it could be effective and helpful for improving clarity."
>
> *Thank you, as stated, we have followed your suggestions to add such summarizing section to the revised manuscript.*
>
> ===
>
> ***Finally, we hope that our answers addressed your concerns well and have improved the manuscript considerably. We would greatly appreciate it if you could consider increasing your rating of this paper, as it matters a lot to us! Thank you very much for your time!***

---

> > ### Author Response · Authors · 2023-11-22
> > **Reminder for Discussion**
> >
> > As the end of the author-reviewer discussion is approaching, we kindly ask for your review of our revised paper and response. If our revisions have adequately addressed your concerns, we'd greatly appreciate your consideration in adjusting the scores. Should you have any remaining questions or require further clarification, please don't hesitate to let us know. We would be more than happy to answer any further questions.

---

### Official Review · Reviewer_iydj · 2023-11-01

**Soundness:** 3 good
**Presentation:** 2 fair
**Contribution:** 2 fair
**Rating:** 5
**Confidence:** 4

**Summary:**

The paper proposes a method for hierarchical planning by learning spatio-temporal abstraction. The approach is tested for zero-shot generalization.

**Strengths:**

- The paper supports the importance of spatial and temporal abstractions together.

- The approach to minimize plans with non-existent checkpoints is interesting.

**Weaknesses:**

- In my opinion, the paper lacks an overall description of the algorithm which largely limits the understanding of the paper.

- A proxy problem is estimated using the models by a conditional generative model that randomly samples checkpoints. These checkpoints are only pruned, not improved and thus the approach seems to be heavily dependent on the plan found earlier even if it can not be achieved at the low level. The delusion suppression approach only minimizes the non-existent checkpoints. Are there any guarantees on whether a checkpoint is achievable?

- There is much research done on learning temporal and/or state abstractions automatically for planning and learning [1,2,3,4]. It is not clear how the proposed work is different or novel compared to these different lines of research. The related work needs a thorough justification of differences with different categories of work.

- The proposed approach seems applicable to grid-like environments only as also included in the experiments.

References:

[1] Hogg, C., Kuter, U. and Munoz-Avila, H., 2009, July. Learning Hierarchical Task Networks for Nondeterministic Planning Domains. In IJCAI (pp. 1708-1714).

[2] Dadvar, M., Nayyar, R.K. and Srivastava, S., 2023, July. Conditional abstraction trees for sample-efficient reinforcement learning. In Uncertainty in Artificial Intelligence (pp. 485-495). PMLR.

[3] Ravindran, B., AC, I. and IIL, R., Hierarchical Reinforcement Learning using Spatio-Temporal Abstractions and Deep Neural Networks.

[4] McGovern, A. and Barto, A.G., 2001. Automatic discovery of subgoals in reinforcement learning using diverse density.

**Questions:**

Included with the limitations.

---

> ### Author Response · Authors · 2023-11-19
> **(1/3) Point-by-point Response to Reviewer iydj's comments**
>
> (1/3)
>
> **Thank you very much for your comments! Here are our point-by-point replies to your concerns.**
>
> * In my opinion, the paper lacks an overall description of the algorithm which largely limits the understanding of the paper.
>
> *We appreciate your comments, and apologize if we hadn't done well enough on the previous attempt: we not only provided a detailed description of each algorithm component, but also an overall framework chart (Fig. 2). In addition, in the appendix, we provided an intuitive overall pseudocode to precisely assemble all components together (whose location we also mentioned in the manuscript. The page limit forced us to compromise on what could be presented in the manuscript and what could be kept in the Appendix). For your convenience, you can access its screenshot here: https://github.com/AnonymousAuthors21/ICLR2024-Rebuttal/blob/main/pseudocode.png*
>
> *We'd also like to point out that Reviewers muKb, 3j7M explicitly remarked positively on the paper's clarity.*
>
> *In response to your concern, we have added more descriptions and references for the readers to quickly navigate to these contents. If there are remaining things that are unclear, please let us know and we’ll add further clarifications. Thank you!*
>
> * A proxy problem is estimated using the models by a conditional generative model that randomly samples checkpoints. These checkpoints are only pruned, not improved and thus the approach seems to be heavily dependent on the plan found earlier even if it can not be achieved at the low level. The delusion suppression approach only minimizes the non-existent checkpoints. Are there any guarantees on whether a checkpoint is achievable?
>
> *We appreciate your comments, but we would like to point out that the proposed checkpoints **do not depend on the earlier plans**, but the previously experienced existent states (not the previous plans) that were reached by the agent and stored in the hindsight replay (off-policy. This means Skipper can learn from data collected by any other agents too). These well-motivated features were explicitly acknowledged by Reviewer 3j7M. We have tried to improve the manuscript on the presentation of this crucial idea for a clearer understanding.*
>
> *The distribution of the generated checkpoints evolves from being skewed to widespread due to the accumulation of more diverse states in the experience replay. Pruning, being a way to improve upon generated proxy problems, have no direct link with the achievement of the checkpoints.*
>
> ***It is the distance estimator that serves the purpose of estimating whether a checkpoint is reachable. If deemed to be unreachable, the planning outcome would not favor choosing such target. We believe that the awareness of the achievability of checkpoints is a distinctive and crucial feature of the Skipper framework.***
>
> *The overall divide-and-conquer strategy of this work is inspired by the difficulty of learning an all-capable policy that can lead to guarantees of achieving a certain checkpoint. And thus, our motivating theory (Theorem 1) provides guarantees on the solution to the overall problem without assuming the checkpoint achievability. In the future work, we will investigate how to train the low-level policy more effectively such that together with a guarantee on the checkpoint achieving policy and the current theory framework, a more insightful theory may be developed.*
>
> *We have again tried our best to improve on the presentation accordingly to your suggestions. Hopefully, this would address your concerns. Please let us know if the approach is clearer now and whether you have further suggestions.*
>
> * There is much research done on learning temporal and/or state abstractions automatically for planning and learning [1,2,3,4]. It is not clear how the proposed work is different or novel compared to these different lines of research. The related work needs a thorough justification of differences with different categories of work.
>
> *We apologize if we hadn’t done a well enough job in such matter. In the submission, Section “Related Works” (right AFTER experiments section), we tried our best to differentiate and justify our method from 5+ aspects and 12+ related works. Now, to improve, we have additionally incorporated your suggested papers in the appropriate locations:*
>
> ***[1]**: The paper’s main contribution is a stochasticity compatible HP framework that does not relate to temporal or spatial abstractions. We have added it to the manuscript in the Appendix where we verified Skipper’s compatibility with stochastic environmental dynamics.*
>
> ***(this reply is to be continued in the next part of the response)***

---

> > ### Author Response · Authors · 2023-11-19
> > **(2/3) Point-by-point Response to Reviewer iydj's comments**
> >
> > (2/3)
> >
> > ***(continuing the last reply)***
> >
> > *[2,3]: The two papers’ main contribution is a special form of state abstraction (not spatial abstraction) that is dynamic state space partitioning, one through conditional abstraction trees and another a state clustering algorithm. Contrastively, our spatial abstraction (not state abstraction) focuses on establishing partial states over partial information of any selected decision timing for the agent to be able to focus on a sub-task.*
> >
> > *These two works above are less related to planning, but more to HRL. The recommended works’ state partitioning ideas are similar to that of the **planning** framework proposed in (Zhang et al., 2021), against which we have compared our method extensively in the related works section. Despite the use of the word “abstraction”, the two works here do not relate to our proposed spatial abstraction which is “dynamically focusing attention on relevant aspects of the state space”. We have added the acknowledgement of these works into the related works section for the comparison between the similarities and differences of the discussed HRL methods. We also added arguments to clarify the definition of spatial abstraction in this paper.*
> >
> >
> > *[4]: This paper contributed to proposing subgoals using by identifying bottlenecks in the state space, with no relevance to spatial and temporal abstractions. We have added acknowledgements in the related works section.*
> >
> > ***While learning abstractions is investigated in the literature, the contribution of this paper is fundamentally different from any existing work to the best of our knowledge.***
> >
> > *For your convenience, here is a copy of the new version of the related works section:*
> >
> > *https://github.com/AnonymousAuthors21/ICLR2024-Rebuttal/blob/main/related_work_updated.png*
> >
> > *While we cannot acknowledge all relevant works in the manuscript, again, we try our best to discuss in detail the related representative works. **Hopefully, after incorporating your suggestions, your concerns regarding the relationship between this work and others can be addressed**. Thank you very much!*
> >
> > * The proposed approach seems applicable to grid-like environments only as also included in the experiments.
> >
> > *We acknowledge the fact that the experiments done for this work are only in the gridworld environments. In the paper, we had to choose a DP-solvable environment to demonstrate the coherence of empirical performance with our motivating theory. Due to the page limit, we could not include more experimental contents in the submission.*
> >
> > *However, despite that we only tested on gridworld experiments, Skipper is not designed to specifically address gridworld problems, and we believe that **there is no reason to deny its applicability on other environments without evidence**. In fact, we are already working on its extension to ProcGen, which is the only visual benchmark that we know that could provide limited insights into the agents’ generalization abilities. The submission is intended to provide a basis for the future extension of its experimental scenarios, and we hope that its own contributions can be recognized.*
> >
> > *We have also added a section dedicated to the weaknesses and limitations for clearer understanding of this work (due to the page limit, it is placed as of now in the Appendix of the revised manuscript, here is a copy):*
> >
> > ***(this reply is to be continued in the next part of the response)***

---

> ### Author Response · Authors · 2023-11-19
> **(3/3) Point-by-point Response to Reviewer iydj's comments**
>
> (3/3)
>
> ***(continuing the last reply)***
>
> > We generate future checkpoints at random by sampling the partial description space. Despite the post-processing such as pruning, the generated checkpoints do not prioritize on the predictable, important states that matter the most to form a meaningful long-term plan. We would like to continue investigating the possibilities along this line;
>
> > The current implementation is intended for pixel input fully-observable tasks with discrete state and action spaces. Such a minimalistic form is because we wish to isolate the unwanted challenges from other factors that are not closely related to the idea of this work, as well as to make the agent as generalist as possible. On the matter of the compatibility with continuous spaces, Skipper is naturally compatible with continuous actions spaces and the only thing we will need to do is to replace the baseline agent with a compatible one such as TD3 (Fujimoto et al., 2018); on the other hand, for continuous state spaces, the identification of the achievement of a checkpoint becomes tricky. This is due to the fact that a strict identity between the current state and the target checkpoint may be ever established, we either must adopt a distance measure for approximate state equivalence, or rely on the equivalence of the partial descriptions (which is adopted in the current implementation). We intentionally designed the partial descriptions to be in the form of bundles of binary variables, so that this comparison could be done fast and trivially for any forms of the state space; for partial observability, despite that no recurrent mechanism has been incorporated in the current implementation, the framework is not incompatible. To implement that, we will need to augment the state encoder with recurrent or memory mechanisms and we need to make  the checkpoint generator directly work over the learned state representations. We acknowledge that future work is needed to verify Skipper’s performance on the popular partially-observable benchmark suites, which requires the incorporation of components to handle partial observability as well as scaling up the architectures for more expressive power;
>
> > We are far from understanding the precise boundaries of the motivating theory on proxy problems, since it only indicates performance guarantees on the condition of estimation accuracy, which in turn does not correspond trivially to a set of well-defined problems. We are eager to explore, outside the scope of sparse-reward navigation, how this approach can be used to facilitate better generalization, and at the same time, try to find more powerful theories that guide us better.*
>
> ====
>
> ***Finally, we hope that our answers addressed your concerns well and we hope that we demonstrated, through our replies, our utmost dedication to this line of research. We would greatly appreciate it if you could consider increasing your rating of this paper, as it matters a lot to us! Thank you very much for your time!***

---

> > ### Author Response · Authors · 2023-11-22
> > **Reminder for Discussion**
> >
> > As the end of the author-reviewer discussion is approaching, we kindly ask for your review of our revised paper and response. If our revisions have adequately addressed your concerns, we'd greatly appreciate your consideration in adjusting the scores. Should you have any remaining questions or require further clarification, please don't hesitate to let us know. We would be more than happy to answer any further questions.

---

> > > ### Comment · Reviewer_iydj · 2023-12-02
> > >
> > > I have read all the reviews and author's responses. I think that the paper in its current form lacks diversity in the domains for sufficient empirical analysis, and would benefit from examples of spatial abstraction as it is not clear if it is just a form of state abstraction or different from it. Other reviewers have also raised these critical points. I am maintaining my score, but happy to discuss further.

---

### Official Review · Reviewer_muKb · 2023-11-01

**Soundness:** 3 good
**Presentation:** 4 excellent
**Contribution:** 3 good
**Rating:** 6
**Confidence:** 3

**Summary:**

The paper proposes a new model-based reinforcement learning agent, Skipper, with the explicit aim to generalize its learned "skills" to novel situations. Skipper creates a "proxy problem" for a given task. Proxy problems are finite graphs created at decision time, whose vertices ("checkpoints") are states and whose directed edges are temporally abstract transitions between these states. The vertices are proposed by a generative model (which splits the state into two parts: a context (which is a general description of the environment), and partial description (which is the agent's current state) , while the estimates of the edges (holding time and expected reward) are learned while interacting with the environment. The training is the sum of the losses of the vertex generation and edge estimation.

The algorithm is evaluated on the MiniGrid-BabyAI framework against the modelfree RL agent that is the basis of the architecture and two relevant Hierarchical Planning methods (LEAP and Director). The evaluation metric is generalization across different instantiations of the environment: subsequent difficulties mean more lava states in the grid. It is demonstrated that Skipper makes use of the context latent representation, as well as the estimated edges to transfer learned policies faster than the baselines.

**Strengths:**

Originality: splitting the state to context and partial information the way it is proposed and using it for transfer across slightly altered tasks seems intuitive yet to the best of my knowledge original.

Quality: experiments ran over 20 seeds, which is substantially more than the usual 5 and is generally recommended the absolute minimum for a pilot study. The plots are well annotated, captioned with error bars included.

Clarity: the paper is well-written and easily understood in most parts.

Significance: the results as presented appear to be significantly better than the baselines. The proposed method coupling state (or "spatial") and temporal abstraction and using it for transfer across tasks is an interesting avenue for further research.

**Weaknesses:**

As mentioned above, the paper has many strengths, however, it is unfortunately not devoid of weaknesses. There are certain parts of the paper that I did not understand, hence I asked several questions in the next section. Furthermore, there is very little information about the way experiments were carried out (with regards to hyperparameter tuning for the proposed algorithm and the baselines on the various environments). These details are required to assess the validity of the claims and to assure reproducibility.

Once my doubts are cleared, I will consider raising my score accordingly.

I listed the experiments ran with 20 seeds as strength and it is. However, in order to claim significance, it is suggested that after the pilot study with 20 seeds, statistical power analysis should be performed to determine the necessary sample size [1][2].

[1] How Many Random Seeds? Statistical Power Analysis in Deep Reinforcement Learning Experiments, Cedric Colas, Olivier Sigaud, and Pierre-Yves Oudeyer, 2018
[2] Empirical Design in Reinforcement Learning, Andrew Pattersion, Samuel Neumann, Martha White, Adam White, 2023

I recommend either adding a statistical power analysis to the paper or softening the claims a little bit.

The following typos/style issues did not affect the score:

The abbreviation RL should be defined as reinforcement learning (RL) the first time it is used., even though it is clear for the audience what it stands for, it is good practice.

s⊙  is defined much later than when first introduced (in Theorem 1) which is confusing.

Section 2, 2nd paragraph, also Section 4 1s paragraph: no need to capitalize the word following a semicolon (;) or end the previous subsentence with a full stop and the capitalize the start of the sentence.

Section 2

1st paragraph: Q: |\mathcal{S}| \times |\mathcal{A}|  (| | missing for S)

2nd paragraph: "evaluated _on_ unseen tasks, _in which_ there are environment variations, but the core strategies need to finish the task remain [...]" (not remains)

3rd paragraph: "Recent work suggests that this approach" -> the latter approach

Figure 2 should be moved at least one page down, there are concepts in it that are explained much later.

"The following estimates are learned with using distributional RL" - with or using

3.3

the expression "on top of" occurs twice and I find it rather ambiguous. Could you use some other expression, e.g. with or after, or anything else that conveys what you are trying to say.

3.3.2

3rd paragraph: "Similarly to $V_{\pi}$, we would want to know the cumulative discount leading to the target s⊙ under $\pi$." - There is no need for this sentence. Just start the next sentence as "The cumulative discount leading to the target s⊙ under $\pi$ is difficult to learn..."

3.4

second paragraph: "tall order" - this is slang, consider changing it to something like "difficult task"

5

"Prior to LEAP (Nasiriany et al., 2019), path planning with evolutionary algorithms was investigated by Nair & Finn (2019); Hafner et al. (2022); Mendonca et al. (2021) propose world models to assist temporally abstract background planning."

Not clear where the first sentence ends and the second begins. Use a full stop and an 'and' where applicable.

**Questions:**

Could you please define spatial abstraction and highlight how it is different from state abstraction? (ideally with a citation) I had a look at all cited papers in the relevant subsection in Section 5, but non of them contained the term "spatial abstraction". Machin et al. 2019 used the term "spatial (and temporal) information", but it was not defined.

“Existing RL agents either operate solely based on intuition (model-free methods) or are limited to reasoning over mostly shortsighted plans (model-based methods)” could you please back this rather strong statement with a citation (especially the first part).

"Model-based HRL agents can be prone to blindly optimizing for objectives without understanding the consequences." - again, at least a citation would be in order for such a claim.

“For any set of options defined on an MDP, the decision process that selects only among those options, executing each to termination, is a Semi-MDP (SMDP, Puterman 2014)” Could you please point to the part in the book that defines SMDPs as such?

“Taking inspiration from conscious information processing in brains [...]” - could you please cite the works whose findings inspired the introduced local perceptive field selector?

3.3.2, the update rule for the Cumulative Reward: could you please explains where is the KL-divergence in that update rule? To me it just looks like each $v(s,a)$ is updated with the TD-target $(r(s,a,s') + \gamma v(s,a))$?

"The following estimates are learned with using distributional RL, where the output of each estimator takes the form of a histogram over scalar support (Dabney et al., 2018)"  - I don't understand what this sentence is trying to say, and the linked paper did not help. Could you please elaborate?

How were hyperparameters tuned for the algorithm (the algorithmic specific ones, such as timeout, pruning threshold, k for k-medoid, etc. as well as the ones basis distributed DDQN) and the various baselines for this task?

---

> ### Author Response · Authors · 2023-11-19
> **(1/4) Point-by-point Response to Reviewer muKb's comments**
>
> (1/4)
>
> **Thank you very much for your comments! Here are our point-by-point replies to your concerns.**
>
> * "I listed the experiments ran with 20 seeds as strength and it is. However, to claim significance, it is suggested that after the pilot study with 20 seeds, statistical power analysis should be performed to determine the necessary sample size [1][2].
> [1] How Many Random Seeds? Statistical Power Analysis in Deep Reinforcement Learning Experiments, Cedric Colas, Olivier Sigaud, and Pierre-Yves Oudeyer, 2018
> [2] Empirical Design in Reinforcement Learning, Andrew Pattersion, Samuel Neumann, Martha White, Adam White, 2023
> I recommend either adding a statistical power analysis to the paper or softening the claims a little bit."
>
> *Thank you very much for your constructive suggestions. We have invested effort to conduct pairwise Welch’s t-tests (for better claims of statistical significance with 20 seed runs) as well as statistical power analyses for all training configurations. The tables below (via urls) show the results (established over the final performance of each run):*
>
> *In the first table, we are showing the pairwise comparison between Skipper-once and the other methods. For each cell of the table, NO is presented if the pairwise t-significance is lower than set threshold (alpha = 0.05) and otherwise, the minimum number of independent seed runs that achieves statistical power (beta < 0.2) is reported. The insignificant results are marked in bold.*
>
> *https://github.com/AnonymousAuthors21/ICLR2024-Rebuttal/blob/main/power_once.png*
>
> *Similarly, here is the table for Skipper-regen v.s. other methods:*
>
> *https://github.com/AnonymousAuthors21/ICLR2024-Rebuttal/blob/main/power_regen.png*
>
> *Based on this new evidence, we have softened our claims of significance accordingly in the revised manuscript (still, for 25, 50, 100 training environments, the results are significant and powerful).*
>
> *Additionally, we have added acknowledging citations to the two suggested references and have added the tables and related discussions to the revised Appendix. Thank you again for the constructive improvement that your comments have brought to this revision.*
>
> * "The abbreviation RL should be defined as reinforcement learning (RL) the first time it is used., even though it is clear for the audience what it stands for, it is good practice."
>
> *Thank you very much for your suggestion. We have adjusted the revised manuscript accordingly.*
>
> * "$S^\odot$ is defined much later than when first introduced (in Theorem 1) which is confusing."
>
> *We are grateful for your suggestion. We have adjusted the revised manuscript accordingly.*
>
> *We added a comment on the notation right after the introduction of checkpoints:*
>
> > The checkpoints are proposed by a generative model and represent a finite subset of states that the agent might experience in the current episode, often denoted as $S^\odot$ in the rest of the paper
>
> * "Section 2, 2nd paragraph, also Section 4 1s paragraph: no need to capitalize the word following a semicolon (;) or end the previous sub-sentence with a full stop and the capitalize the start of the sentence."
>
> *Thank you very much for your suggestion. We have adjusted the revised manuscript accordingly.*
>
> *We addressed the capitalization problem not only in the suggested paragraphs but also checked globally that such problems no longer exist.*
>
> * "Section 2 1st paragraph: Q: |\mathcal{S}| \times |\mathcal{A}| (| | missing for S)"
>
> *Thank you very much for your suggestion. We have adjusted the revised manuscript accordingly.*
>
> *We removed | | from A, since Q should be a mapping from the cross of two spaces, the | | for A was a typo.*
>
> * "2nd paragraph: "evaluated on unseen tasks, in which there are environment variations, but the core strategies need to finish the task remain [...]" (not remains)"
>
> *Thank you very much for your suggestion. We have adjusted the revised manuscript accordingly.*
>
> * "3rd paragraph: "Recent work suggests that this approach" -> the latter approach"
>
> *We appreciate your suggestion. We have adjusted the revised manuscript accordingly.*
>
> * "Figure 2 should be moved at least one page down, there are concepts in it that are explained much later."
>
> *Addressed by moving the figure location according to the suggestion.*
>
> * " 'The following estimates are learned with using distributional RL' - with or using"
>
> *Thank you very much for your suggestion. We have adjusted the revised manuscript accordingly.*

---

> ### Author Response · Authors · 2023-11-19
> **(2/4) Point-by-point Response to Reviewer muKb's comments**
>
> (2/4)
>
> * "the expression "on top of" occurs twice and I find it rather ambiguous. Could you use some other expression, e.g. with or after, or anything else that conveys what you are trying to say."
>
> *Thank you very much for your suggestion. We have adjusted the revised manuscript accordingly.*
>
> *Occurrence regarding the usage of the partial state extractor, changed to:*
>
> > Through $\sigma$, the auxiliary estimators, to be discussed soon, force the bottleneck mechanism to promote aspects relevant to the local estimation of connections between the checkpoints.
>
> *Occurrence regarding the usage of partial state representation, changed to:*
>
> > The rewards and discounts are then estimated from the partial state $\sigma(S)$, based on the agent's behavior.
>
> *Occurrence regarding the addition of a loss signal, changed to:*
>
> > This is implemented trivially by adding a loss to the original training loss for the distance estimator, which we give a $0.25$ scaling for stability.
>
> *Also changed the occurrences in the Appendix.*
>
> * "3.3.2 3rd paragraph: "Similarly to ��, we would want to know the cumulative discount leading to the target s⊙ under �." - There is no need for this sentence. Just start the next sentence as 'The cumulative discount leading to the target s⊙ under � is difficult to learn...'"
>
> *Thank you very much for your suggestion. We have adjusted the revised manuscript accordingly.*
>
> *Changed to:*
>
> > The cumulative discount leading to the target $s_\odot$ under $\pi$ is unfortunately more difficult to learn than $V_\pi$, since the prediction would be heavily skewed towards $1$ if $\gamma \approx 1$.
>
> * "3.4 second paragraph: "tall order" - this is slang, consider changing it to something like 'difficult task' "
>
> *Thank you very much for your suggestion. We have adjusted the revised manuscript accordingly.*
>
> *Changed to “challenge”.*
>
> * "5  'Prior to LEAP (Nasiriany et al., 2019), path planning with evolutionary algorithms was investigated by Nair & Finn (2019); Hafner et al. (2022); Mendonca et al. (2021) propose world models to assist temporally abstract background planning.'
> Not clear where the first sentence ends and the second begins. Use a full stop and an 'and' where applicable."
>
> *Thank you very much for your suggestion. We have adjusted the revised manuscript accordingly.*
>
> *Replaced the ; with a full stop and a “Moreover”.*
>
> * "Could you please define spatial abstraction and highlight how it is different from state abstraction? (ideally with a citation) I had a look at all cited papers in the relevant subsection in Section 5, but none of them contained the term "spatial abstraction". Machin et al. 2019 used the term "spatial (and temporal) information", but it was not defined."
>
> *We wanted to present the narrative that “spatial abstraction” resembles the consciousness information processing in the first sense (selective dynamic attention towards environmental entities, Dehaene et al., 2017). Most existing idea of state abstraction (not spatial) in RL is to simply learn a capable state representation that could handle an overall RL task. While “spatial abstraction” in our presentation focuses on the dynamic partial selection upon an already abstracted full state (which you may view as another layer of (meta-)abstraction over a state representation, which is an abstraction itself over the information of the interaction history with the environment), to make the agent learn to focus on the relevant aspects for the duration of the current option (checkpoint transition). This is like Zhao et al.’s NeurIPS 2021 work on learning a “bottleneck (partial) state” that depends on an already abstracted full state.*
>
> *The full state contains environmental entities that could be used for decision-making of all time steps should be preserved but only some aspects are used in certain decision points. “Spatial abstraction” is a product of both acknowledging the abstraction needed additionally over a full state representation, as well as our take on partial states for options, with a name that coheres accordingly to the temporal abstraction, another key ingredient of Skipper.*
>
> *We have tried to condense the statements above into a footnote in the manuscript to enhance the presentation:*
>
> > We use "spatial abstraction" in this work to denote specifically the attentive behavior of constraining decision-making to partial environmental factors that matter during an option.
>
> *We understand that this is a newly-introduced notion which has not yet gained consensus in the community, hence we are open to suggestions for the improving its presentation.*

---

> ### Author Response · Authors · 2023-11-19
> **(3/4) Point-by-point Response to Reviewer muKb's comments**
>
> (3/4)
>
> * " “Existing RL agents either operate solely based on intuition (model-free methods) or are limited to reasoning over mostly shortsighted plans (model-based methods)” could you please back this rather strong statement with a citation (especially the first part)."
>
> *In this argument, we adopted the categorization of model-free v.s. model-based methods. Model-free methods, which rely on operating on a policy or value estimator act on intuition (we were trying to use the common terminology of behavioral psychology. For example, in Daniel Kahneman’s works on system-1 characteristics.). While in the second half of the argument, we tried to acknowledge the limitations of model-based RL in their temporally extended reasoning abilities (resembling system 2). To the best of our knowledge, most of the model-based agents indeed cannot reason over long-horizon, due to the curse of error accumulation and exploding outcomes with the increment of the simulation depth, hence we used the adjective “shortsighted”. To tone down the argument a little bit, we changed this to “relatively shortsighted”*
>
> *We feel that simply giving a citation here wouldn’t really help for a better understanding from the readers therefore we have added a footnote that condensed the explanation above, with proper citations included:*
>
> > In the view of \cite{daniel2017thinking}, model-free agents act on System-1 intuitions while model-based agents combine System-2 reasoning on top of intuitions.
>
> * " "Model-based HRL agents can be prone to blindly optimizing for objectives without understanding the consequences." - again, at least a citation would be in order for such a claim."
>
> *Thank you very much for your suggestion. We have adjusted the revised manuscript accordingly.*
>
> *Now the argument is backed up with two existing works that discuss such concerns.*
>
> *“Model-based HRL agents can be prone to blindly optimizing for objectives without understanding
> the consequences (Langosco et al., 2022; Paolo et al., 2022).”*
>
> * "“For any set of options defined on an MDP, the decision process that selects only among those options, executing each to termination, is a Semi-MDP (SMDP, Puterman 2014)” Could you please point to the part in the book that defines SMDPs as such?"
>
> *Our sincere apologies. For this part we have forgotten to add a citation to Sutton et al. 1999. We would like to thank the reviewer for pointing this out. We have added the necessary citation. This statement is the same as Theorem 1 in Sutton et al. 1999. We have added the citation to Puterman 2014 to cite SMDPs rather than the whole sentence.*
>
> * "“Taking inspiration from conscious information processing in brains [...]” - could you please cite the works whose findings inspired the introduced local perceptive field selector?"
>
> *Thank you very much for your suggestion. We have adjusted the revised manuscript accordingly.*
>
> *Added a citation to a survey that summarizes the attention-related aspects of conscious information processing (Dehaene et al.,2020).*
>
> * "3.3.2, the update rule for the Cumulative Reward: could you please explains where is the KL-divergence in that update rule? To me it just looks like each is updated with the TD-target? The following estimates are learned with using distributional RL, where the output of each estimator takes the form of a histogram over scalar support (Dabney et al., 2018)" - I don't understand what this sentence is trying to say, and the linked paper did not help. Could you please elaborate?"
>
> *TD targets are not conflicting with a distributional RL approach, in fact these two are quite orthogonal.*
>
> *Dabney et al., 2018 proposed the following way to use TD style updates to (provably) learn the distribution of the discounted return in RL: Instead of letting the value estimator output a scalar, we ask it to output a histogram (taking a softmax over a vector output) over a support of scalars. We regress the histogram towards the TD target, where such target is converted into a skewed histogram towards which KL-divergence can be used to accomplish the estimation.*
>
> *In our statement that you quoted above, we were referring to the just-explained technique specifically, because of the existence of other possible quantization methods that could also achieve the learning of a distribution of the return. Each of our estimators learns the distribution of the return of a specific reward function.*
>
> *Due to the page limit, we could not include a detailed description of the methods used for distributional learning in the main manuscript. However, with your suggestions, we have added related descriptions in the Appendix. Please let us know if this answer addressed your concerns, if not, we are open to adding or changing arguments to improve the presentation.*

---

> ### Author Response · Authors · 2023-11-19
> **(4/4) Point-by-point Response to Reviewer muKb's comments**
>
> (4/4)
>
> * "How were hyperparameters tuned for the algorithm (the algorithmic specific ones, such as timeout, pruning threshold, k for k-medoid, etc. as well as the ones basis distributed DDQN) and the various baselines for this task?"
>
> ***The following blocks are condensed and added to the revised manuscript:***
>
> ***Timeout and Pruning Threshold***
>
> > Intuitively, we set the timeout to be equal to the distance pruning threshold. The timeout kicks in when the agent thinks a checkpoint can be achieved within e.g. 8 steps, but already spent 8 steps yet still cannot achieve it.
>
> > This leads to how we tuned the pruning (distance) threshold: we fully used the advantage of our experiments on a fully DP-solvable environment: with a snapshot of the agent during its training, we can sample many <starting state, target state> pairs and calculate the ground truth distance between the pair, as well as the failure rate of reaching from the starting state to the target state given the current policy \pi, which is again enabled by DP. Then, we plot them as the x and y values respectively for visualization. We found such curves to evolve from flat high failure rate with all distances at the beginning, to a monotonically increasing curve (w.r.t. the true distances), where at small true distances, the failure rates are near zero. We picked 8 because the curve starts to grow explosively when the true distances are more than 9.
>
> *These visualizations were key validations for the effectiveness of our proposed estimators (we do the same validations for other estimators as well). Our architectures (depths, widths of MLPs, size of distributional outputs) on top of the baseline distributional DDQN were iterated (minimally) to achieve desired results on these visualizations while minimizing the computational demand.*
>
> *We compiled a gif for you to visualize how the error in the estimation of the reward estimator evolve during training, which can be accessed here:*
>
> *https://github.com/AnonymousAuthors21/ICLR2024-Rebuttal/blob/main/evolution_reward_error.gif*
>
> ***K for k medoids***
>
> *We tuned this by running a sensitivity analysis on skipper agents with different k’s. The results are in the Appendix B.6.*
>
> *Additionally, we prune from 32 checkpoints because 32 checkpoints could achieve an ok coverage of the state space (maximally 12^2=144 states) as well as its friendliness to NVIDIA accelerators (batch size 32 is fast).*
>
> ***Size of local Perception Field***
>
> *We used a local perception field of size 8 because our baseline model-free agent would be able to solve and generalize well within an 8x8 task but not larger. Roughly speaking, our spatial abstraction breaks down the overall tasks into 8x8 sub-tasks, which the policy could comfortably solve.*
>
> *The local perception field is implemented in its simplest form for vision-based tasks: an 8x8 convolutional layer (no padding) on top of the embedded input.*
>
> ***Model-free Baseline Architecture***
>
> *The baseline architecture (distributional, Double DQN) was heavily influenced by the architecture used in (Zhao et al., 2021), which demonstrated success on similar but smaller-scale experiments (8x8). The difference is that while they used computationally heavy components such as transformer layers on a set-based representation, we replaced them with a simpler and effective local perception component, as discussed just now. We validated successfully our model-free baseline performance on the tasks proposed in (Zhao et al., 2021).*
>
> *No training-related hyperparameters were changed in this work, which include the intervals of training, the learning rate, the optimizer, etc. These are in fact identical to those in Rainbow (Hessel et al., 2017).*
>
> ===
>
> ***Finally, we hope that our answers addressed your concerns well and have improved the manuscript considerably. We would greatly appreciate it if you could consider increasing your rating of this paper, as it matters a lot to us! Thank you very much for your time!***

---

> > ### Author Response · Authors · 2023-11-22
> > **Reminder for Discussion**
> >
> > As the end of the author-reviewer discussion is approaching, we kindly ask for your review of our revised paper and response. If our revisions have adequately addressed your concerns, we'd greatly appreciate your consideration in adjusting the scores. Should you have any remaining questions or require further clarification, please don't hesitate to let us know. We would be more than happy to answer any further questions.

---

> > > ### Comment · Reviewer_muKb · 2023-11-23
> > >
> > > Dear Authors,
> > >
> > > I will need more time to revisit the revised version of the paper and the details in the response. I will update my score according to the information given. Thank you for your response and the updates to the paper.

---

> > > > ### Author Response · Authors · 2023-11-23
> > > > **Thank you and**
> > > >
> > > > Thank you for your reply. As the official timeline states that the author-reviewer discussion time ends in about 7.5 hours, we may not be able to answer your new questions. Hopefully the point-to-point responses to your raised concerns would address the weaknesses of the submission.
> > > >
> > > > Sincerely,
> > > > Authors

---

### Author Response · Authors · 2023-11-19
**Overall Response to all the Reviewers**

Thank you very much for your constructive suggestions. In the revision, we have mainly worked on the following points:

 * Improved the clarity and the presentation of the paper

* Added more rigorous statistical significance tests as well as statistical power test for more precise claims of performance advantages.

* Added more details about hyperparameter / architecture tuning.

* Added a discussion section on the assumptions and limitations of the paper.

* Added more related works to differentiate ours to existing ones.

* Added more discussion about the theory

* Updated some figures

Thank you again and we hope these would address your concerns. **The point-by-point response to each of your raised concerns will follow.**

---

### Meta-Review · Area_Chair_WyPR · 2024-01-10

**Metareview:**

This submission proposes a new model-based reinforcement learning method that can generalize learned skills for sparse reward setups. The method uses a "proxy" graph to spatially and temporally abstract the problem into a high-level "proxy" problem and generalize within the abstracted space. It is shown to be quite effective on a navigation-oriented gridworld problem.

The suggested method is a novel approach of generalizing skills, bringing a new insight to the community. Also, the experiments were thoroughly ran with 20 different seeds with good performance gain. Thus, the submission is suggested for "accept". However, the review committee (including the AC) raises a common concern of its limitation on the scenarios (e.g. grid world). While the authors rebutted that there is no fundamental barrier for the method to extend to general scenarios (and also they are running experiments in an academic setup), the exact challenges and its extendability is not tested at all in the current draft. There is also a lot of unclarity on how to exactly generate a proxy graph with different domains of problems. Thus, the impact of the method in its current form is limited and the method is to be tested by others. So, the stronger recommendation (spotlight or oral) couldn't be given.

Thus, some suggestions to consider are:
- More evaluations would make the proposed submission much more solid and understand its full capacity.
- Related works. It is missing quite a bit of skill-related related works:
* Hausman et al. “Learning an embedding space for transferable robot skills” ICLR 2018
* Bagaria and Konidaris “Option discovery using deep skill chaining” ICLR 2020
* Pertsch et al. “Accelerating Reinforcement Learning with Learned Skill Priors” CoRL 2020
* Xie et al. “Latent Skill Planning for Exploration and Transfer” ICLR 2021
* Dalal et al. “Accelerating Robotic Reinforcement Learning via Parameterized Action Primitives” NeurIPS 2021
* Shi et al. “Skill-based Model-based Reinforcement learning” CoRL 2022
* Nasiriany et al. “Augmenting Reinforcement Learning with Behavior Primitives for Diverse Manipulation Tasks” ICRA 2022

**Justification For Why Not Higher Score:**

see above

**Justification For Why Not Lower Score:**

n/a

---

### Decision · Program_Chairs · 2024-01-16

Accept (poster)